# Developmental mRNA m$^5$C landscape and regulatory innovations of massive m$^5$C modification of maternal mRNAs in animals

Jianheng Liu [1,6], Tao Huang[1,6], Wanying Chen[1,6], Chenhui Ding[2,6], Tianxuan Zhao[1,6], Xueni Zhao[1,6], Bing Cai[2,6], Yusen Zhang[1,6], Song Li[2], Ling Zhang[1], Maoguang Xue[3,4], Xiuju He [1], Wanzhong Ge [3,4,5✉], Canquan Zhou [2✉], Yanwen Xu [2✉] & Rui Zhang [1✉]

m$^5$C is one of the longest-known RNA modifications, however, its developmental dynamics, functions, and evolution in mRNAs remain largely unknown. Here, we generate quantitative mRNA m$^5$C maps at different stages of development in 6 vertebrate and invertebrate species and find convergent and unexpected massive methylation of maternal mRNAs mediated by NSUN2 and NSUN6. Using *Drosophila* as a model, we reveal that embryos lacking maternal mRNA m$^5$C undergo cell cycle delays and fail to timely initiate maternal-to-zygotic transition, implying the functional importance of maternal mRNA m$^5$C. From invertebrates to the lineage leading to humans, two waves of m$^5$C regulatory innovations are observed: higher animals gain cis-directed NSUN2-mediated m$^5$C sites at the 5' end of the mRNAs, accompanied by the emergence of more structured 5'UTR regions; humans gain thousands of trans-directed NSUN6-mediated m$^5$C sites enriched in genes regulating the mitotic cell cycle. Collectively, our studies highlight the existence and regulatory innovations of a mechanism of early embryonic development and provide key resources for elucidating the role of mRNA m$^5$C in biology and disease.

[1] MOE Key Laboratory of Gene Function and Regulation, Guangdong Province Key Laboratory of Pharmaceutical Functional Genes, State Key Laboratory of Biocontrol, School of Life Sciences, Sun Yat-Sen University, Guangzhou, PR China. [2] Guangdong Provincial Key Laboratory of Reproductive Medicine, Center for Reproductive Medicine and Department of Gynecology & Obstetrics, the First Affiliated Hospital, Sun Yat-Sen University, Guangzhou, PR China. [3] Division of Human Reproduction and Developmental Genetics, Zhejiang Provincial Key Laboratory of Precision Diagnosis and Therapy for Major Gynecological Diseases, Women's Hospital, Zhejiang University School of Medicine, Hangzhou, Zhejiang 310006, China. [4] Institute of Genetics, Zhejiang University School of Medicine, Hangzhou, Zhejiang 310058, China. [5] Cancer Center, Zhejiang University, Hangzhou, Zhejiang 310058, China. [6]These authors contributed equally: Jianheng Liu, Tao Huang, Wanying Chen, Chenhui Ding, Tianxuan Zhao, Xueni Zhao, Bing Cai, Yusen Zhang. ✉email: wanzhongge@zju.edu.cn; zhoucanquan@mail.sysu.edu.cn; xuyanwen@mail.sysu.edu.cn; zhangrui3@mail.sysu.edu.cn

RNAs contain over a hundred different modifications, and such modifications have been recently recognized as important gene regulatory features[1–5]. The conservation and evolution of RNA modification enzymes have been extensively studied (e.g refs. [6,7].). Moreover, the distribution of RNA modification sites in noncoding RNA species, such as tRNA and rRNAs, have been described in different species[1,8,9]. Recent technical advances have also revealed a number of mRNA modifications. However, except for m6A methylation and adenosine-to-inosine RNA editing[9–12], the in vivo functions and evolution of most types of mRNA modifications have not been fully investigated. m5C, mediated by DNMT2 and the NSUN methyltransferase family, is a well-known RNA modification. Previous studies have shown that m5C is present in diverse RNA species[13]. The functions and regulation of m5C in tRNAs and rRNAs have been extensively characterized[14–18]. Moreover, the functional relevance of m5C in mRNAs has been recently emerged[19–22].

Initially m5C was reported to be widespread in mRNAs[23–25], but it was later recognized that many sites that had been initially identified were likely spurious[26,27]. Accurate and systematic transcriptome-wide detection of mRNA m5C has been challenging, thus we still have no clear and consistent view on the abundance and whereabouts of this modification. To overcome this challenge, we developed a framework to robustly identify and quantify mRNA m5C sites based on RNA bisulfite sequencing (BS-seq) of enriched mRNAs[28,29]. With this approach, we typically identified only several hundred mRNA m5C sites in a given adult tissue in mammals. We and others also revealed that these limited number of sites are classified as two types: Type I m5C sites, which contain a downstream G-rich triplet motif and locate at the 5' end of hairpin structures, are methylated by NSUN2[29,30]; Type II m5C sites, which contain a downstream UCCA motif and locate in the loops of hairpin structures, are methylated by NSUN6[31–33].

To understand the landscape, function, and evolution of mRNA m5C, we sequenced samples from 6 animal species spanning 800 million years of evolution to construct quantitative maps of mRNA m5C at different stages of development. Unexpectedly, we observed mRNA m5C as a specialized modification that is largely restricted to maternal mRNAs. We further used cell models and animal models to investigate the mechanism underlying the extensive methylation of maternal mRNAs and the biological importance of m5C in early embryonic development. Finally, we applied comparative epitranscriptomic approaches to reveal two major m5C regulatory innovation steps and the rapid evolution of individual m5C sites.

## Results

### Maternal mRNAs are methylated to an unprecedented extent in both vertebrate and invertebrate species

To reveal the m5C profiles during development in both vertebrate and invertebrate species, we conducted mRNA BS-seq for a total of 29 zebrafish and 34 Drosophila melanogaster (D. mel) samples covering all developmental stages (Fig. 1a, Supplementary Data 1). For each stage, two replicates were profiled, and high-confidence m5C sites methylated at levels ≥10% were called as we previously described[29]. The methylation levels were highly consistent between replicates, supporting the accuracy of m5C level quantification (Supplementary Fig. 1). Surprisingly, 6,259 and 8,974 exonic m5C sites were identified in 3,187 and 5,021 mRNAs in early embryonic stages of zebrafish and D. mel, respectively, which were mainly transcribed from the maternal genome (Fig. 1b, Supplementary Data 2). The number of m5C sites dropped dramatically after the maternal-to-zygotic transition

(MZT) and remained low throughout the rest of developmental stages (Fig. 1b, Supplementary Fig. 2). Gini coefficient analysis[29] indicated that these maternal mRNA m5C sites were unlikely to be false positives (Supplementary Fig. 3a, Methods), and the decrease in the overall m5C level during development was also confirmed using dot blotting (Supplementary Fig. 3b).

Inspired by this finding, we next asked whether other animals, especially mammals, have the same striking epitranscriptomic pattern. We collected 19 samples from four more species (humans, mice, X. laevis, and X. tropicalis) that represent two major vertebrate classes[34], with a focus on oocyte and early embryo samples (Fig. 1a, Supplementary Data 1). Two technical replicates were profiled for each sample except for the difficult-to-collect mammalian oocyte and early embryo samples. We further confirmed that robust results could be obtained with low RNA input (oocyte and early embryo samples in humans and mice) using our method (Supplementary Fig. 4). Intriguingly, we found that such an unusual pattern was indeed conserved in other animals (Fig. 1b, Supplementary Fig. 2, Supplementary Fig. 3a). For example, 32,941 sites were observed in 8,592 mRNAs in human Metaphase II (MII) oocytes. Besides, the massive mRNA methylation was restricted to oocytes, and only a low-level of methylation was present in the surrounding follicle cells (Supplementary Fig. 2). Additionally, the numbers and methylation levels of m5C sites seemed to be gradually increased during oocyte maturation and fertilization (Fig. 1b). Quantitatively, the densities of m5C sites in maternal mRNAs were 10~30 times higher than those in later developmental stages or adult tissues (Fig. 1b). Moreover, the densities of m5C sites were found to be continuously increased from invertebrates to the lineage leading to mammals (Fig. 1b). Combined, we identified a convergent maternal mRNA-specialized massive m5C methylation.

### Maternal mRNA m5C is mainly deposited by NSUN2 in non-human species

We sought to determine the writer proteins of maternal mRNA m5C. Since the m5C pattern in maternal mRNAs greatly differed from that in zygotic mRNAs, we asked whether other unknown methyltransferases might mediate maternal mRNA m5C methylation or whether these m5C sites were also deposited by the two known methyltransferases. We utilized MEME[35] to call motifs among these maternal mRNA m5C sites (Methods). In all species, all of the top enriched motifs can be classified as Type I- or Type II-like motifs (Supplementary Data 3). Since the sequence motif alone is a robust signature[31], we grouped maternal mRNA m5C sites based on their motifs and examined their structural features. In all species, sites with the two motifs were located at the 5' end or in the loops of hairpin structures (Supplementary Fig. 5a), consistent with the known structural features of Type I and Type II sites, respectively. Concordantly, we found that NSUN2 and NSUN6 were highly conserved among the species we analyzed (Supplementary Fig. 5b, c). Taken together, these data indicate that maternal mRNA m5C is deposited by NSUN2 and NSUN6.

Next, we characterized the landscape of m5C sites in maternal mRNAs. We found that a small fraction of maternal m5C sites were present in zygotic mRNAs (Supplementary Fig. 5d). Overall, for D. mel and zebrafish, ~92% of maternal m5C sites were located in CDS regions; however, for frogs and mammals, maternal m5C sites in UTRs took up 20.6% to 36.8% of the sites (Fig. 1c). In most species, Type I sites were the major type of maternal mRNA m5C, accounting for 86% to 97% of the sites; intriguingly, a substantial fraction (9664, 23.7%) of sites in humans were Type II sites (Fig. 1c). When looking into the m5C distribution along the transcripts, we found that Type I sites in mammals had a 5' end enrichment, in agreement with previous

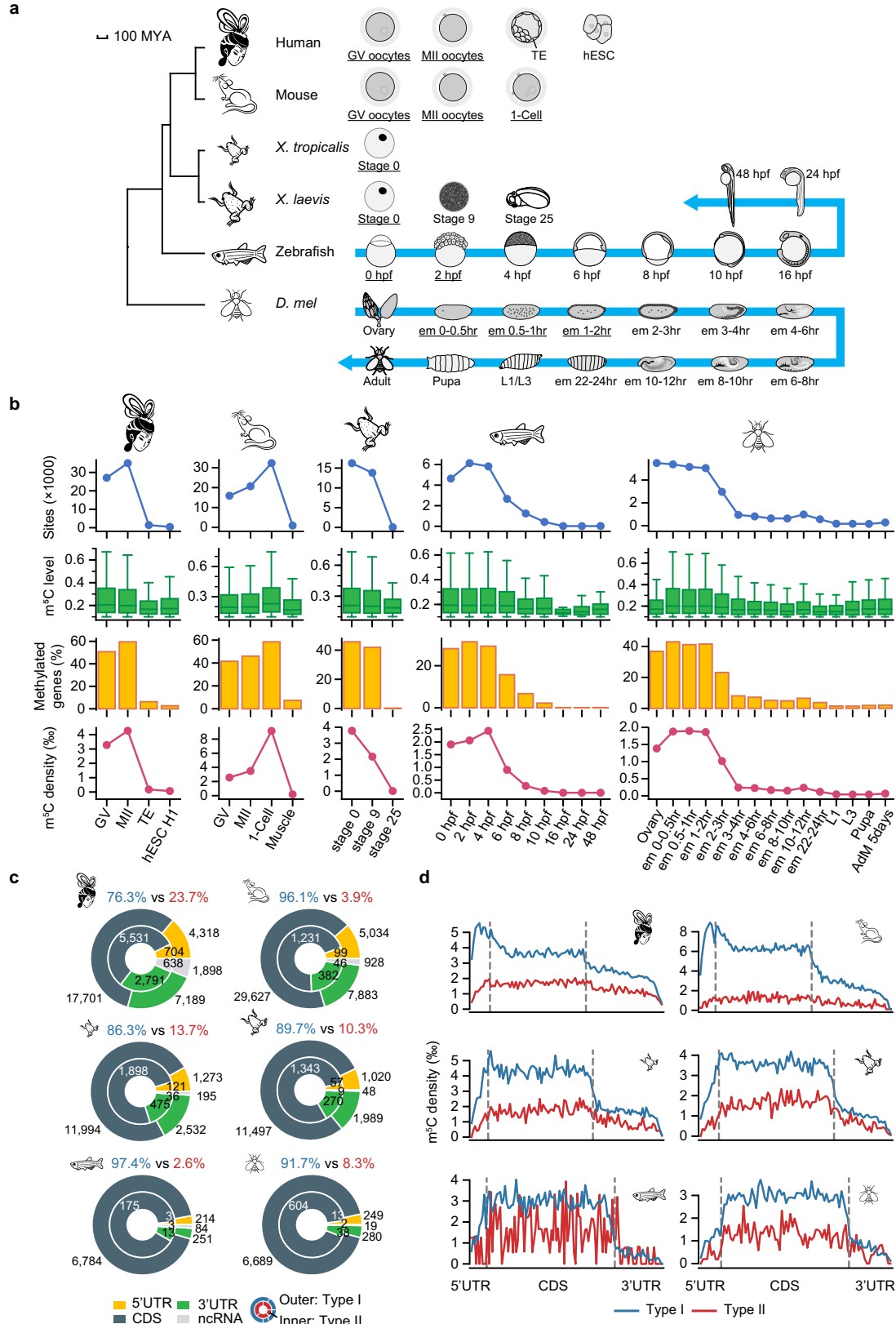

observations[28]; however, this pattern did not hold in the other species we examined (Fig. 1d, Supplementary Fig. 5e). In all species, Type I and Type II sites had a similar distribution in CDS and 3'UTR regions (Fig. 1d, Supplementary Fig. 5e).

To understand the possible mechanisms underlying the maternal mRNA methylation, we focused on the writer protein

of Type I sites, which were the major type of maternal mRNA m5C. Using published RNA-seq data, we found that expression levels of NSUN2 could not fully explain the mRNA m5C pattern we observed (Supplementary Fig. 6), although it did have the highest expression level in early embryonic stages in zebrafish and frogs. This result suggests the existence of additional mechanisms.

**Fig. 1 Maternal mRNAs are methylated to an unprecedented extent in both vertebrate and invertebrate species by NSUN2 and NSUN6. a** Phylogenic tree (left) and sampled developmental stages of the species in this study (right). The mammals separated from frogs, zebrafish, and *D. mel* about 352, 435, and 797 mya, respectively[34]. The divergence between humans and mice is estimated to be 90 mya and the divergence of the two frog species is estimated to be 57 mya[34]. Stages in which mRNAs were largely transcribed from the maternal genome are underlined. For humans and mice: GV germinal vesicle, MII Metaphase II, TE trophectoderm, hESC human embryonic stem cell, 1-cell 1-cell embryo. For zebrafish: hpf, hours post-fertilization. For *D. mel*: em embryo, L larva, hr hour. Zebrafish embryo images were adapted from Webb et al., 2006;[63] adult zebrafish images were adapted from Kimmel et al.,1995;[64] fly ovary image was adapted from AVILéS et al., 2018;[65] other fly sample images were adapted from Wolpert et al., 2015[66]. **b** The number and methylation level of m⁵C sites, the percentage of transcribed genes with m⁵C sites, and the density of m⁵C sites in the samples we profiled. The density was defined as the number of m⁵C sites per thousand Cs that are covered by at least 20 reads. The mouse muscle sample that has the highest m⁵C level in adult tissues[29] was selected to represent the zygotic mRNA m⁵C pattern. AdM, adult male; H1, hESC H1 line. Boxplots: 25th to 75th percentiles (boxes), medians (horizonal lines), and 1.5 times of the interquartile range (whiskers). The number of data points are provided in Source Data. Numbers of biological replicates are provided in Supplementary Fig. 1. **c** Genic locations of maternal m⁵C sites in each species. The number of m⁵C sites and the percentages of Type I and Type II sites are indicated. **d** The distribution of maternal m⁵C sites along the transcripts in different species. In this analysis, each m⁵C site was binned and the m⁵C density of each bin was calculated (Methods). Bin numbers were based on average lengths among transcripts in different species. Bin numbers (5'UTR:CDS:3'UTR): humans, 10:50:40; mice, 10:60:50; *X. tropicalis*, 10:60:40; *X. laevis*, 10:60:30; zebrafish, 10:80:30; *D. mel*, 10:60:20. Source data are provided as a Source Data file.

There are two unique features of maternal mRNAs: first, they are remarkably stable; second, they are mainly translationally silenced and stored in the cytoplasm[36–38]. Thus, we hypothesized that the location of NSUN2, which determines the duration of its interaction with mRNAs, might be linked to the massive methylation. In *D. mel*, NSUN2 mainly localized in the nucleus of prophase I (PI) oocytes (Fig. 2a, Supplementary Fig. 7a). With the nuclear envelope breakdown, NSUN2 was strongly expressed in the cytoplasm of metaphase I (MI) oocytes (Fig. 2a). Unlike *D. mel*, NSUN2 localized in both the nucleus and the cytoplasm of PI oocytes in humans and mice (Fig. 2b, Supplementary Fig. 7b, c). Similarly, with the nuclear envelope breakdown, NSUN2 was expressed in the cytoplasm of human and mouse MI/MII oocytes (Fig. 2b, Supplementary Fig. 7c). Together with the observation that m⁵C levels were gradually increased during oocyte maturation and fertilization, we proposed that cytoplasmically located NSUN2 protein may persistently interact with maternal mRNAs and lead to progressive addition of m⁵C. Notably, although mammals had lower oocyte NSUN2 expression than lower animals (Supplementary Fig. 6), they still had an extraordinarily high density of maternal mRNA m⁵C. Interestingly, mammalian NSUN2 proteins evolved to be present in the cytoplasm at an earlier stage of oocyte maturation, which might result in a longer interaction time between NSUN2 and maternal mRNAs and complement the lower NSUN2 expression to achieve high methylation. Finally, we mimicked the status of MI/MII oocytes, i.e., persistent contact between translationally silenced mRNAs and NSUN2 in the cytoplasm, by treating HeLa cells with nocodazole to reduce mRNA translation and arrest the cells at the prometaphase (see Methods). Indeed, the treatment led to a gradual and sharp increase of densities and methylation levels of Type I sites (Fig. 2c–e) but not Type II sites (Supplementary Fig. 7d–f), without affecting NSUN2 protein levels (Supplementary Fig. 7g), supporting our hypothesis.

**The removal of Type I sites in *D. mel* leads to a developmental delay.** The observed extensive methylation of maternal mRNAs suggests that m⁵C methylation may play a role in the regulation of maternal mRNAs. A previous study in zebrafish found that YBX1 preferentially bound m⁵C transcripts and that YBX1 interference led to early embryonic development defects[39]. However, as YBX1 has been linked to a wide range of processes that seem unrelated to m⁵C[40], the data that directly reflect the functional significance of mRNA m⁵C and methyltransferase is still missing. Since we found that NSUN2 is the major mRNA m⁵C writer in *D. mel* (Fig. 1c), we examined NSUN2 loss of function in *D. mel* to understand the role of RNA m⁵C in embryogenesis. We first generated an NSUN2 knockout *D. mel* line (Supplementary Fig. 8) and confirmed that the vast majority of maternal mRNA m⁵C sites (96%) were methylated by NSUN2 (Fig. 2f). To characterize the maternal function of NSUN2, we crossed female NSUN2 mutant flies with male wild-type flies to produce maternal NSUN2 knockout embryos. On average, ~35% of fertilized eggs from NSUN2 mutant mothers failed to hatch into first instar larvae (Supplementary Fig. 9a and Supplementary Discussion). Additionally, wild-type eggs typically became larvae at ~21.5 h, however, in mutants, hatching was delayed by ~2 h (Supplementary Fig. 9b and Supplementary Discussion). To specifically examine the consequence of NSUN2 ablation on early embryonic mitotic divisions, we measured the progression through early embryonic cycles by counting the percentage of laid embryos that were in different stages of embryogenesis (Supplementary Fig. 9c). The embryos from NSUN2 mutant mothers showed slower progression through embryogenesis than wild-type flies in two independent experiments (Fig. 2g and Supplementary Fig. 9d). This slower progression was partially rescued by the introduction of the wild-type, but not the catalytically inactive, NSUN2 transgene (Fig. 2g, Supplementary Fig. 9e), and the rescue of m⁵C was confirmed by BS-seq in ovaries (Supplementary Fig. 9f). The same defect was observed in another, independently generated NSUN2 null mutant (Supplementary Fig. 8 and Supplementary Fig. 9g). These observations indicate that maternal NSUN2 is necessary for the normal progression of the cell cycle during early embryogenesis (Supplementary Discussion).

To further verify the observed developmental delay at the molecular level, we profiled wild-type and maternal NSUN2 knockout embryos at different time points using mRNA-seq. Five time points in wild-type embryos and three matched time points in mutant embryos were examined (Supplementary Data 4). Principal component analysis (PCA) showed a transition from one stage to another through developmental time (Fig. 2h). The genes that contributed the most to the first PC were either low at early time periods and increased towards the end of the time course or the inverse (Supplementary Fig. 10a). Thus, PC1 can be considered as a measurement of the progression of development. The PC1 values of mutant embryos (2–3 h and 4–5 h) were smaller than those of wild-type embryos (Fig. 2h), consistent with the observed phenotype of developmental delay. Then, the zygotic transcripts were inferred using transcriptome data from wild-type embryos (Supplementary Fig. 10b), and their expression in mutant flies was assessed. Delayed initiation of a substantial fraction of these transcripts was observed (Supplementary Fig. 10c), in agreement with the phenotype. Taken together, our

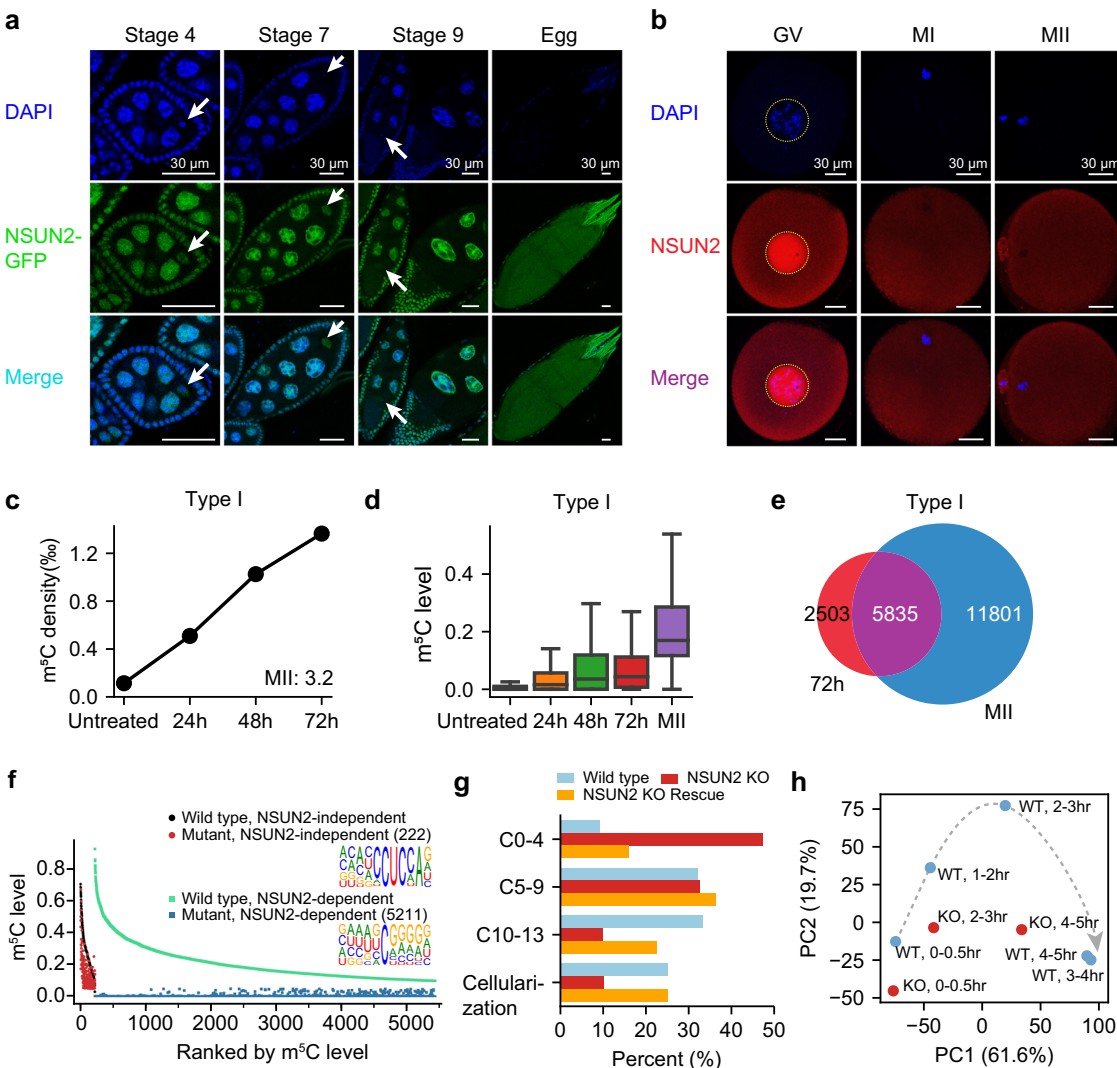

**Fig. 2 The deposition of mRNA m$^5$C in oocytes and the in vivo functions of NSUN2 in early embryogenesis. a** Representative images of egg chambers at different stages for *D. mel* expressing a GFP-NSUN2 fusion protein under the control of its native promoter. DNA was stained with DAPI (blue). Arrows indicate the germ cell that will become the oocyte. The localization of NSUN2 was consistent in all samples of each stage ($n = 12, 14, 17, 20$). Scale bars, 30 μm. **b** Representative images of human GV/MI/MII oocytes stained with anti-NSUN2 antibody. 11 GV, 14 MI, and 14 MII oocytes were stained, and the NSUN2 localization was consistent in all oocytes examined. Scale bars, 30 μm. **c, d** The densities (**c**) and levels (**d**) of Type I m$^5$C sites in HeLa cells treated with nocodazole for 0, 24, 48, and 72 h (1 sample per time point). In panel d, a union of sites with levels ≥10% in at least one sample was used for analysis ($n = 5039$). The density of mRNA m$^5$C sites in MII oocytes is indicated in **c**. Boxplots: 25th to 75th percentiles (boxes), medians (horizonal lines), and 1.5 times of the interquartile range (whiskers). **e** Overlaps of Type I mRNA m$^5$C sites between HeLa cells treated with nocodazole for 72 h and MII oocytes. **f** Comparison of m$^5$C methylation levels in the ovaries between wild-type flies and NSUN2 knockout flies. Type I and Type II m$^5$C sites were shown separately. **g** Quantification of the percentage of 0–2 h embryos that were in different stages of embryogenesis. Embryos were grouped into four stages: within four cleavage cycles, 5–9 cycles, 10–13 cycles, cellularization (as shown in Supplementary Fig. 9c). The first independent experiment on mutant #1 line was shown. The numbers of flies used are provided in Source Data. **h** PCA showing the first two PCs, which together explain 82.6% of the variance in the transcriptome data. The amount of variance explained by each PC is indicated on each axis. Wild-type and maternal NSUN2 knockout embryos are colored blue and red, respectively. Source data are provided as a Source Data file.

results and the previous study based on mRNA m$^5$C reader protein YBX1 complement each other to suggest the functional significance of maternal mRNA m$^5$C.

**Cis-regulatory innovation leads to the gain of 5' end Type I sites in higher animals.** It is known that the earliest periods of animal development are the most divergent in terms of both gene expression and protein sequence evolution[41]. Having established the functional significance of maternal mRNA m$^5$C sites, we next examined the degrees of evolutionary change in their abundances,

genic locations, and targeted genes to ask whether they may contribute to the divergence in early embryonic development processes between species. By analyzing the genic locations and distributions of Type I and Type II sites, two major innovations were observed. The first innovation is a transition from depletion to enrichment of 5' end Type I sites from invertebrates to the lineage leading to mammals. To address the driving force of this innovation, we first tested whether it might be due to trans-regulatory changes in NSUN2 protein in mammals. Since human maternal and zygotic Type I sites showed similar 5' end pre-ferences, we expressed human, mouse, and zebrafish NSUN2

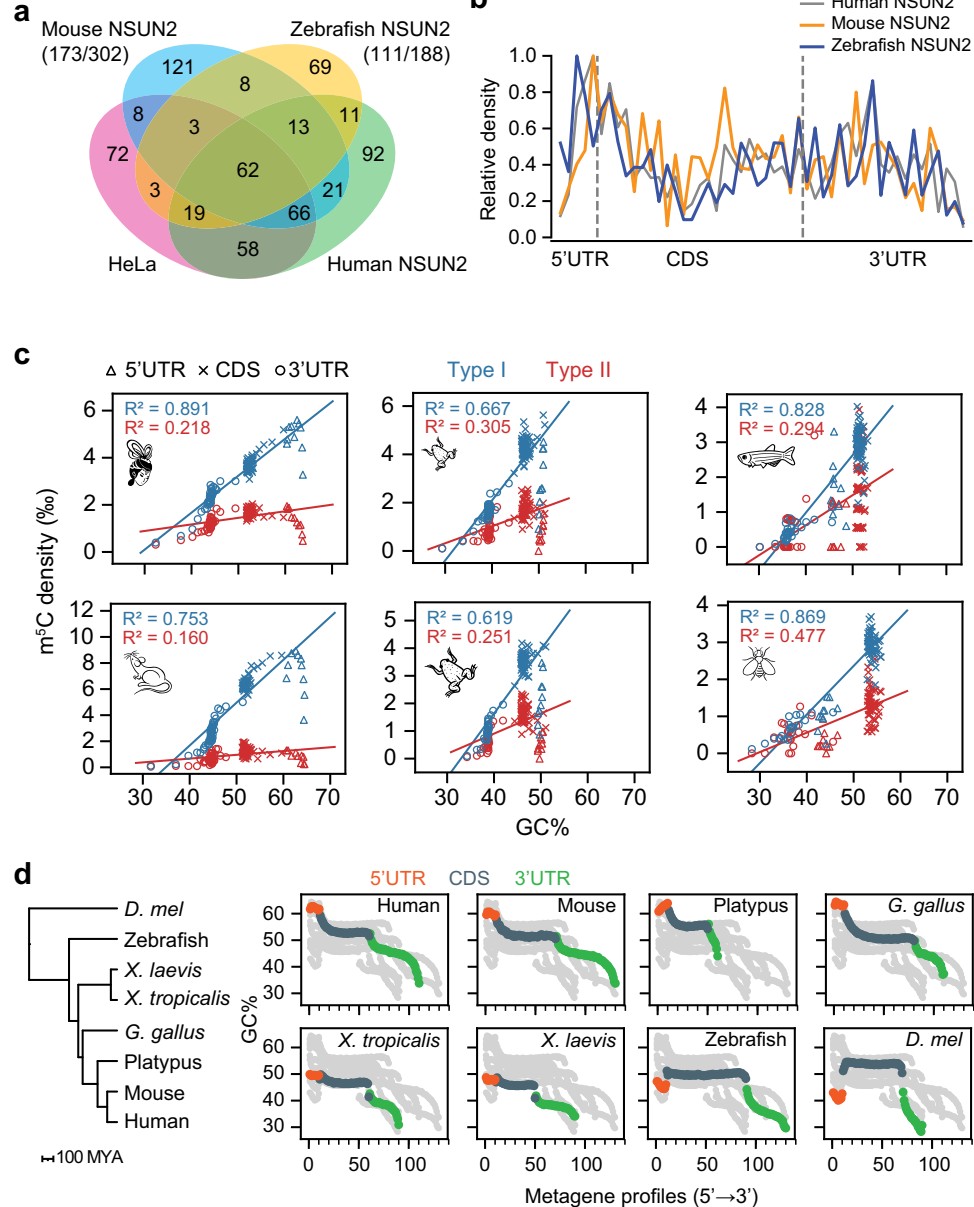

**Fig. 3 Cis-regulatory innovation leads to the gain of 5' end Type I sites in higher animals. a** Venn diagram showing the overlaps of Type I sites identified in wild-type HeLa cells and NSUN2 knockout HeLa cells expressing human, mouse, and zebrafish NSUN2 individually. Only sites that are covered by at least 20 reads in all samples were analyzed. The number of Type I sites overlapped with HeLa or human NSUN2 against the total number of Type I sites methylated by mouse or zebrafish NSUN2 was shown in parentheses. **b** The distribution of Type I $m^5C$ sites in NSUN2 knockout HeLa cells expressing human, mouse, and zebrafish NSUN2 individually. The density was first calculated as in Fig. 1d and then normalized to the bin with the highest density. **c** The relationship between maternal $m^5C$ density and GC content of the transcriptome. In this analysis, each maternal $m^5C$ site was binned as in Fig. 1d and the $m^5C$ density and GC content of each bin was calculated and plotted. Type I and Type II sites were calculated separately and Pearson correlation coefficient of determination is indicated. **d** Metagenomic analysis of GC contents in different species. With the investigation of more representative vertebrate species, we found that the 5' end high GC content was obtained in the common ancestor of birds and mammals. Source data are provided as a Source Data file.

genes individually in NSUN2 knockout HeLa cells (expressing zygotic mRNAs) that lacked endogenous Type I sites to test their methylation activities. We found that all homologous NSUN2 proteins behaved similarly to human NSUN2 protein: (1) over 50% of the sites methylated by mouse and zebrafish NSUN2 overlapped with human NSUN2 sites (Fig. 3a), although their methylation levels varied (Supplementary Fig. 11a); (2) sites methylated by mouse and zebrafish NSUN2 showed sequence contexts and structural preferences similar to those methylated by human NSUN2 (Supplementary Fig. 11b, c); and (3) both mouse

and zebrafish NSUN2 proteins exhibited a 5' end methylation preference (Fig. 3b), although zebrafish NSUN2 methylated only a small number of cytosines at the 5' end of its own transcriptome (Fig. 1c, d). Thus, trans-regulatory changes in NSUN2 proteins are unlikely to result in such innovation.

This motivates us to evaluate the contribution of cis-regulatory elements to the acquisition of 5' end Type I sites. We found that, at intraspecies level, the densities of Type I but not Type II $m^5C$ sites were highly correlated with the local GC contents (Fig. 3c). At interspecies level, there was an increase in the GC content at

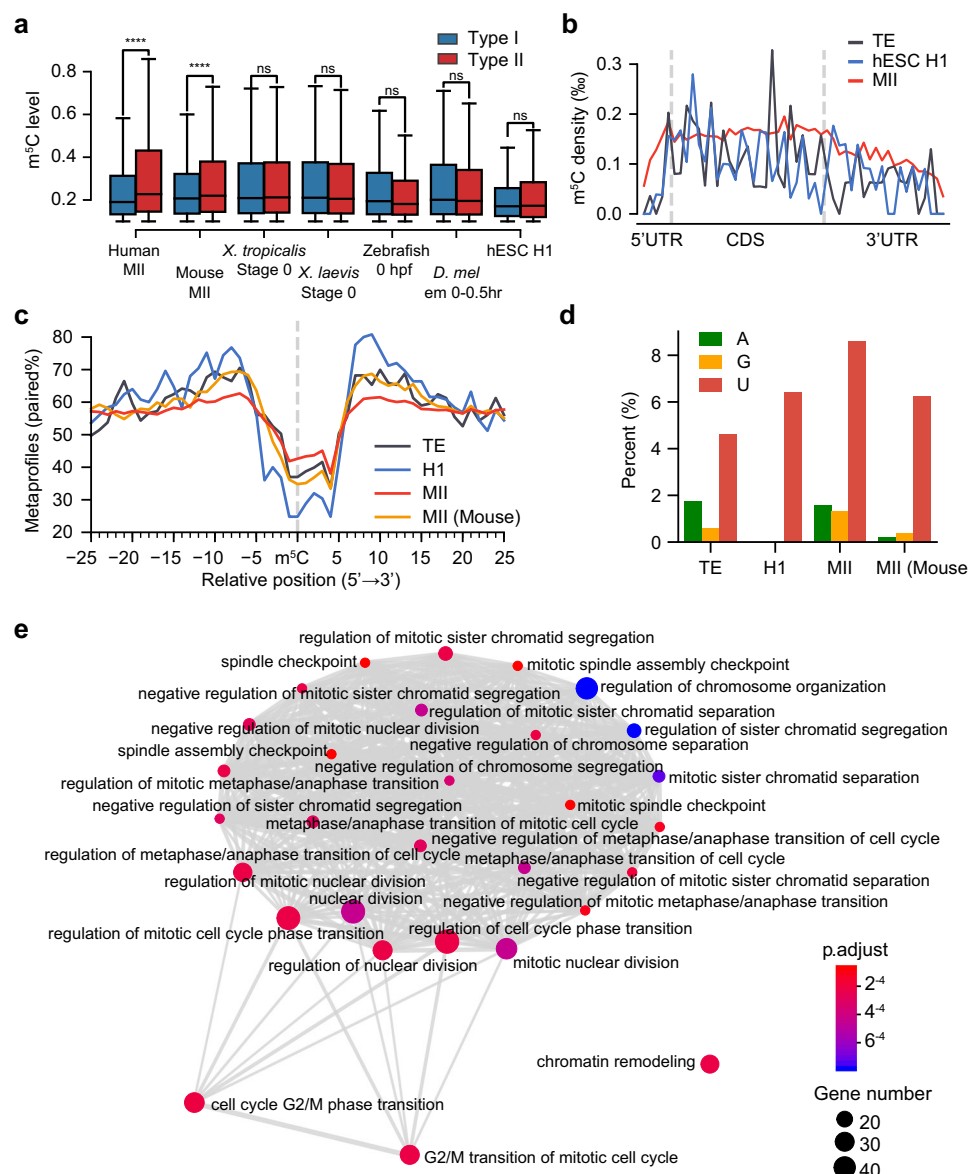

**Fig. 4 Trans-regulatory innovation leads to the gain of thousands of human-specific Type II sites. a** Comparison of the levels of Type I and Type II sites in selected samples that represent maternal mRNA methylation status in different species and in hESC H1 that represents human zygotic mRNA methylation status. Note that in mouse oocyte samples, only 3.9% of the sites were Type II sites. Boxplots: 25th to 75th percentiles (boxes), medians (horizonal lines), and 1.5 times of the interquartile range (whiskers). The *P*-values were calculated using one-sided Student's *t*-test. ****$p$ < 0.0001; ns, not significant. From left to right: Type I sites: 26625, 19652, 15994, 14554, 4557, 4907, 312; Type II sites: 8388, 1042, 2530, 1679, 138, 482, 125; *P*-values: 2.61e-151, 0.184, 1.18e-6, 0.527, 0.931, 0.93, 0.914. **b** The distribution of Type II m⁵C sites in human MII oocytes, hESC H1, and human TE. The density was first calculated as in Fig. 1d and then normalized to the bin with the highest density. **c** The metaprofiles of the secondary structure of Type II m⁵C sites and flanking regions in human MII oocytes, hESC H1, human TE, and mouse MII oocytes. **d** Base composition at position +3 of the core motif for Type II sites in human MII oocytes, hESC H1, human TE, and mouse MII oocytes. **e** Enrichment map plot (see Methods) for genes that were only regulated by NSUN6 in human MII oocytes. The top 30 terms were shown. Source data are provided as a Source Data file.

the 5' end of transcripts from lower animals to higher animals (Fig. 3d), which was coupled with the gradual acquisition of 5' end Type I sites and consistent with the requirement of the structure-dependent m⁵C methylation. These results highlight a cross-talk between gene structure evolution and epitranscriptomic innovation that gradually emerged in higher animals.

**Trans-regulatory innovation leads to the gain of thousands of human-specific Type II sites.** The second innovation is the gain of several thousand Type II sites in humans. In addition to their increase in number and proportion, human Type II sites had a significantly higher methylation level than Type I sites (Fig. 4a).

This difference was not found in other species or in human zygotic mRNAs (Fig. 4a). The human maternal Type II sites were distributed evenly along the transcripts, similar to the case in other species and in human zygotic mRNAs (Fig. 4b), suggesting that the gain of maternal Type II sites was not due to cis-directed evolution in humans. Closer inspection of the sequence and structural features revealed a less stringent requirement of secondary structure (Fig. 4c, Supplementary Fig. 11d) and more non-cytosine bases at position +3 (Fig. 4d) in human maternal Type II sites than in human zygotic sites and sites from other species (mouse as one example). This result indicates that NSUN6 evolved to specifically broaden its methylation target selection for

human maternal mRNA regulation. In line with this, we found that mimicking the status of MI/MII oocytes using HeLa cells could not dramatically increase the densities and methylation levels of Type II sites (Supplementary Fig. 7d–f). This result might be explained by the oocyte-specific regulation of NSUN6, the existence of oocyte-specific factors, such as RNA binding proteins, to cooperate with NSUN6 to promote methylation, or the presence of a negative regulator that represses Type II site methylation in HeLa cells. Notably, we also found that humans evolved to have a relatively higher NSUN6 expression in oocytes and early embryonic stages (Supplementary Fig. 6). These data together suggest that the expansion of Type II sites in humans is likely due to trans-regulatory innovation in NSUN6 expression and methylation activity. The expansion of Type II sites in humans led to ~1100 genes that only be regulated by NSUN6. Intriguingly, these genes were highly enriched in mitotic cell cycle regulation (Fig. 4e). Thus, NSUN6-mediated methylation may act as a new layer of cell division regulation in early embryonic development in humans.

**The surprisingly rapid evolution of individual m5C sites**. The ample data in multiple species provided us the opportunity to examine the evolution of individual m5C sites. We found that the number and percentage of conserved sites dropped rapidly as the divergence time increased (Fig. 5a). For example, ~8.36% and 16.75% of the conserved cytosines were methylated in human-mouse pair and *X. laevis-X. tropicalis* pair in spite of variable methylation levels between species (Supplementary Fig. 12a); however, only 3.34% and 2.25% of the conserved cytosines were methylated between humans and zebrafish and humans and *X. tropicalis*, respectively. At the gene level, we also only observed weak cross-species conservation of methylation site numbers, as exemplified in human-mouse pair and *X. laevis-X. tropicalis* pair (Supplementary Fig. 12b). Moreover, the overall methylation levels of individual genes were not conserved between species (Supplementary Fig. 12c). Given the surprisingly rapid evolution of individual m5C sites, we focused on the more closely related human-mouse pair and *X. tropicalis-X. laevis* pair to investigate the contribution of motifs and structures to methylation divergence. An examination of motif conservation revealed that mutations introduced in the motif regions were associated with the loss of methylation in 55% and 37% of the nonconserved m5C sites in human-mouse pair and *X. tropicalis-X. laevis* pair, respectively. The conserved m5C sites shared a strong stem-loop structure (Fig. 5b, Supplementary Fig. 13a). In contrast, sites that were methylated in one species but not in the other had a much weaker stem-loop structure in the species with the loss of methylation (Fig. 5b, Supplementary Fig. 13a). Thus, both motif and structure changes contributed substantially to the evolution of m5C methylation. Interestingly, the conserved Type II sites in humans had a less stringent stem-loop structure requirement than those in mice, echoing the finding that human NSUN6 evolved to broaden its methylation target selection by including a greater variety of structures.

We further applied a logistic generalized linear model (GLM) (see Methods) to quantitively evaluate the contribution of sequence and structural features for the evolution of m5C sites. For Type I sites, the gain of Gs at positions +1 to +4 of the core motif was positively correlated with the gain of methylation, while the gain of Us, particularly at position +1, contributed most to the loss of methylation (Fig. 5c, Supplementary Fig. 13b). Moreover, the downstream base-pairing status, particularly at position +1, was important in determining the gain or loss of methylation (Fig. 5c, Supplementary Fig. 13b). For Type II sites, any mutations in the core motif, except at position +3, led to a

loss of methylation (Fig. 5d, Supplementary Fig. 13c), consistent with our previous experimental verification[31]. In addition, the loss of a pairing site in the stem and the gain of a pairing site in the loop were overall negatively correlated with the gain of an m5C site, although the contribution of sites at different positions varied (Fig. 5e, Supplementary Fig. 13d). Finally, to provide experimental support for the inference of Type I site evolution, we performed a high-throughput mutagenesis assay for 5 Type I substrates[31] (Supplementary Fig. 13e, Supplementary Data 5). We found that the number of Gs at the 3' G-rich motif was positively correlated with methylation, confirming the strong requirement for this motif (Fig. 5f). Moreover, the stem, especially the five base pairs downstream of the m5C site, was essential for methylation, and a single mutation in this region greatly interrupted methylation (Fig. 5g). These results suggest that the methylation potential of m5C sites is encoded in the surrounding sequence motif and the stem-loop structure, and the gain and loss of individual sites during evolution is largely predicable.

## Discussion

In summary, our work afforded an unprecedented view of the dynamic landscape of mRNA m5C in animals. We found convergent and unexpected massive methylation of maternal mRNAs, suggesting that m5C methylation may have evolved a specialized role in labeling and regulating maternal mRNAs. We further proposed a possible mechanism underlying the extensive methylation of maternal mRNAs. Using *D. mel* as a model, we revealed that NSUN2 knockout embryos undergo cell cycle arrest or delay and fail to initiate the MZT in a timely manner. Notably, a limitation of our study is that we were unable to determine if the observed effects of NSUN2 loss of function on embryogenesis are due to changes in mRNAs or tRNAs. A previous study in zebrafish found that YBX1 preferentially bound m5C-containing mRNAs and YBX1 interference led to early gastrulation defects, suggesting that maternal mRNA m5C might play a role in early embryonic development[39]. However, because YBX1 has been linked to a wide range of processes that are unrelated to m5C, the data that directly reflects the functional significance of mRNA m5C and methyltransferase is still missing. As our study identified the major methyltransferase responsible for the maternal mRNA m5C methylation in *D. mel* and observed a similar phenotype upon its knockout, the previous YBX1 study and our study complement each other to suggest the functional significance of maternal mRNA m5C. Maternal mRNAs are known to be tightly controlled by parallel mechanisms[42–44]. Thus, mRNA m5C-mediated regulation may coordinate with other known maternal mRNA regulators to regulate maternal mRNAs.

Comparative epitranscriptomic analyses revealed two striking innovation steps from invertebrates to mammals and then humans. First, the emergence of more structured 5' end regions in mammals was accompanied with the gain of NSUN2-mediated m5C sites at the 5' end of the mRNAs. Structured RNA elements in the 5'UTR play essential roles in translational regulation[45], thus the gain of m5C in this region in higher animals might represent the addition of a new layer of regulation to control translational initiation[30]. However, the most surprising and striking innovation of m5C is observed in humans. Human NSUN6 protein acquired enhanced activity by relaxing its structural requirements, as well as motif requirements, to specifically achieve maternal mRNA methylation (Supplementary Fig. 14). This innovation gave rise to thousands of Type II m5C sites that are enriched in genes regulating the mitotic cell cycles and might contribute to the distinctive feature of human preimplantation development[46,47]. Note that

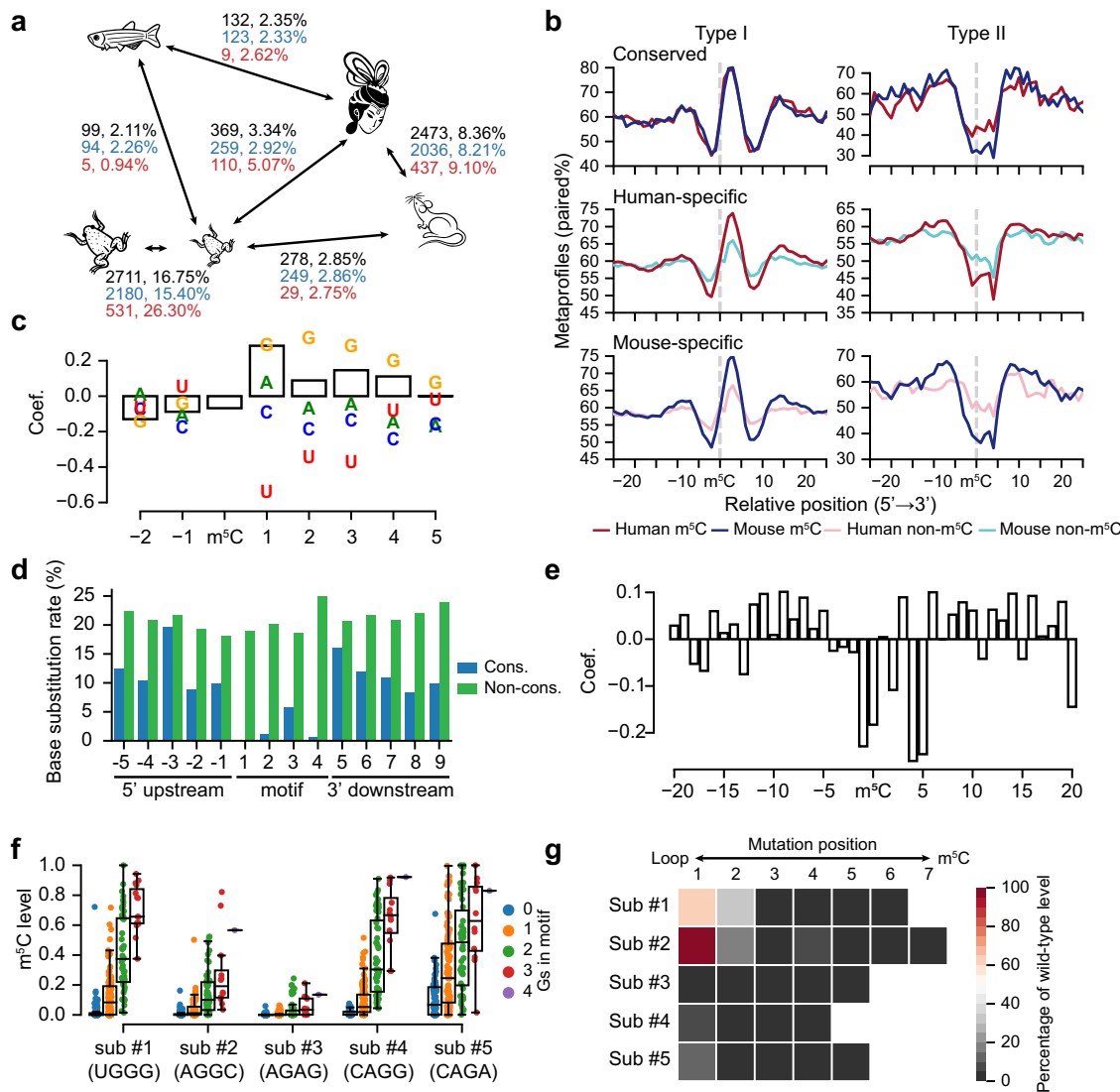

**Fig. 5 Sequence- and structure- dependent evolution of mRNA m⁵C.** **a** Pairwise comparison showing the numbers and percentages of conserved m⁵C sites between vertebrate species. The lengths of lines are proportional to the divergence times between species. The numbers and percentages of all m⁵C sites (black), Type I m⁵C sites (blue), and Type II m⁵C sites (red) were shown separately. Samples used: humans and mice, MII oocytes; frogs, stage 0; zebrafish, 0 hpf; fly, 0–0.5 h embryos. Conserved methylation sites, m⁵C sites with a level ≥10 % in one species and with a level >5% in another species. Only conserved Cs covered by at least ten reads in both species were considered. **b** The metaprofiles of the secondary structure for conserved and non-conserved Type I and Type II m⁵C sites between humans and mice. Conserved, m⁵C sites that were methylated in both species. Human-specific or mouse-specific, m⁵C sites that were Cs at the DNA level in both species but only methylated in humans or mice. Samples used: human and mouse MII oocytes. **c** GLM coefficients of sequence and structural factors for the gain or loss of Type I sites between humans and mice (see Methods). Coefficients of the bases were shown in characters and coefficients of the structures were shown in bars. **d** Base substitution rates in the motif and flanking regions of conserved and non-conserved Type II sites between humans and mice. **e** GLM coefficients of structural features for the gain or loss of Type II sites between humans and mice. Bars on the positive *Y*-axis indicate that the paired status of that base is a positive factor for methylation. **f** The relationship between the numbers of Gs in the motifs of five Type I substrates and their m⁵C levels. All 256 types of motifs were tested. For each motif, the base-pairing status was maintained by the introduction of compensatory mutation. Sub, substrates. Boxplots: 25th to 75th percentiles (boxes), median (horizonal line), and 1.5 times of the interquartile range (whiskers). Numbers of data points are provided in Source Data. **g** The relationship between point mutations in the stems of 5 Type I substrates and their m⁵C levels. The position of each mutation in 5 Type I substrates was shown in Supplementary Fig. 13e. Source data are provided as a Source Data file.

although the sequence and structural features suggest that Type II sites in maternal mRNAs were likely NSUN targets, future studies based on knockout or knockdown models are needed to confirm NSUN6 as the writer protein of Type II sites in maternal mRNAs and investigate its functional role, in particular in humans. Our discovery demonstrates that, in parallel with epigenetic marks[48–50], epitranscriptomic marks are a new source of evolutionary innovation during animal development.

## Methods

**Ethical statement**. All mouse experiments were carried out according to guidelines of Institutional Animal Care and Use Committee (IACUC) of Sun Yat-Sen University.

Informed consent for human subject research: The study was approved by the Ethical Committee of the First Affiliated Hospital of Sun Yat-sen University (Approval Reference Number: 2019-467). All procedures followed were in accordance with the ethical standards of the Ethical Committee of the First Affiliated Hospital of Sun Yat-sen University and with the Helsinki Declaration of 1975, as revised in 2000 (5). A total of 501 immature oocytes, including 246 GV,

14 MI, and 241 MII, were donated from 148 women, who were with an age between 20 and 39 years old and undergoing Intracytoplasmic Sperm Injection (ICSI). Sixty-seven blastocysts were donated from 37 patients, who were with an age between 26 and 36 years old, undergoing preimplantation genetic test for monogenic diseases (PGT-M), and tested as euploid but affected with beta-thalassemia. All oocytes and embryos were obtained with written informed consent signed by the donor couples. The informed consent confirmed that the couple donors were voluntarily donating oocytes and embryos for research on the mechanisms of human oocyte maturation and early embryonic development with no financial payment.

**_Drosophila_ genetics and sample collection.** _D. mel_ W1118 was used as a wild-type control. The mutant alleles for NSUN2 were generated using the CRISPR/Cas9-induced mutagenesis system following the previously described procedure[51]. In brief, a sgRNA sequence (TGGCCGAGCTGCACCAGA) was designed using flyCRISPR Optimal Target Finder (http://www.flyrnai.org/crispr/). The sgRNA template oligonucleotide was synthesized and cloned into the donor vector pUAST-attB. The sgRNA-pUAST-attB plasmid was microinjected into 86Fb (Bloomington 24,749) embryos. The microinjection was performed by the Core Facility of Drosophila Resource and Technology, SIBCB (CAS). The sgRNA transgenic line was crossed with vas-Cas9, so the progeny expressed both sgRNA and vas-induced Cas9 in the germline. The genotype of F1 was screened by Sanger sequencing. Two mutants were selected for the experiments.

The UAS full-length NSUN2 (UAS-dNSUN2+ cDNA) flies were obtained from a previous study[52]. Daughterless-GAL4 was used to drive the re-expression of NSUN2 in the ovaries. The reported Daughterless-GAL4 expression in the ovaries was confirmed by crossing with UAS-GFP flies.

To obtain the mRNA m5C profiles across the entire developmental stages in _D. mel_, 14 stages were selected and obtained as follows. For the embryo collection, ten stages, including 0–0.5, 0.5-1, 1–2, 2–3, 3–4, 4–6, 6–8, 8–10, 10–12, and 22–24 h, were collected. Later staged sample collections started with synchronized embryos and included resynchronizing with appropriate age indicators. Two larval stages (L1 (43–44 h after egg-lay) and L3 (84–85 h after egg-lay)), 1 pupal stage (3 days after L3), 1 adult sexed stage (5 days males), and ovaries (virgins fed with yeast for 3 days) were collected. 500–1000 embryos, ~50 larvae, ~50 pupae, ~50 ovaries, and ~50 adult flies were used for RNA isolation in each stage.

**Zebrafish maintenance and sample collection.** Samples were obtained from natural crosses of wild-type AB zebrafish. Embryos were reared at 28.5 °C and all experiments were performed as close to this temperature as possible[53]. Nine stages, including 0, 2, 4, 6, 8, 10, 16, 24, and 48 hpf were collected for RNA BS-seq. Approximately 200 embryos and ~150 larvae were used for RNA isolation in each stage. For adult tissue collection, 6-month-old zebrafish was used. For each tissue type, samples from three individuals were pooled for RNA isolation.

**_X. laevis_ and _X. tropicalis_ embryo collection.** Wild-type _X. laevis_ from a stock maintained by the Xenopus Resource Center, Hangzhou, China was used. Three stages (stage 0, stage 9, and stage 25) were collected and ~200 embryos were used for RNA isolation in each stage. Wild-type _X. tropicalis_ was used, and ~200 stage 0 embryos were used for RNA isolation.

**Mouse oocyte and one-cell embryo collection.** Oocytes were obtained from the ovaries or oviducts of 7–10-week-old C57BL/6 J female mice. GV oocytes were harvested from the minced ovaries 48 h after the mice were administered 10 IU of pregnant mares' serum gonadotrophin (PMSG, Solarbio, P9970) intraperitoneally. The GV oocytes were cultured in G-IVF (Vitrolife) for 4–6 h, and then MI oocytes were collected. To obtain MII oocytes, mice were administered 10 IU human chorionic gonadotrophin (hCG, Univ, 230734) intraperitoneally, 44–48 h after injection of 10 IU of PMSG (Solarbio, P9970). Twelve h later, the MII oocytes were picked under a microscope by fallopian tube dissection. One-cell embryos were collected from 7 to 8-week-old C57BL/6 J females mated with C57BL/6 J males, and ovulation was induced as described above. The embryos were flushed from the reproductive tract 24 h after hCG administration. For all samples collected, the zona pellucida (ZP) was removed by treatment of 10 IU/ml Hyaluronidase (Sigma, H6254).

**Human oocyte and early embryo collection.** After oocyte retrieval, oocyte-cumulus complexes (OCCs) were incubated for 4–6 h in an equilibrated G-IVF medium (Vitrolife) supplemented with 10% human serum albumin (HSA) (Vitrolife). Next, all OCCs were denuded using 80 IU/ml Hyaluronidase solution (Sage). GV oocytes were vitrified for cryopreservation, while MI oocytes were in vitro matured. In brief, MI oocytes were matured in G-IVF medium supplemented with 10% HSA for 20–24 h after oocyte denudation. Matured MII oocytes were determined by the presence of the first polar body. After in vitro maturation, MII oocytes were vitrified for cryopreservation. For RNA isolation, cryopreserved oocytes were warmed and treated with 0.5% (w/v) pronase (Sigma-Aldrich, P-8811) in G-MOPS medium (Vitrolife) supplemented with 5% HSA to remove the ZP and remnant of cumulus cells. After ZP removal, ZP-free oocytes were washed

thoroughly in PBS containing 0.1% bovine serum albumin (BSA) (Sigma, Cat#A1933) before lysis. The remnant of cumulus cells and the first polar bodies were removed by careful pipetting.

The TE of day 5 blastocysts was isolated mechanically based on microsurgical technique with the aid of an inverted microscope (Olympus IX71) equipped with micromanipulators (Eppendorf Transferman NK2) and an infrared diode laser (Octax Laser Shot System). In brief, day 5 blastocysts were immobilized by a holding pipette (TPC). The biopsy pipette was inserted into the blastocoel, and the inner cell mass (ICM) cells were sucked in and gently pulled out with laser assistance. The rest of blastocysts were cut as trophoblast cells clumps by laser and mechanical pulling force.

**mRNA-seq.** Wild-type or NSUN2 mutant virgins were crossed with wild-type males for 2 days in advance. For each time period, 70–100 embryos were collected to ensure RNA yield and that samples were representative. Total RNA was isolated with TRIzol reagent and Direct-zol RNA MiniPrep kit (Zymo research). For each sample, 1 μg total RNA was used for library construction. Polyadenylated RNA was separated from total RNA using Oligo dT Magnetic Beads (Vazyme). RNA was then used for library construction, following the manufacturer's protocol (NEBNext Ultra II Directional RNA Library Prep Kit, NEB). Libraries were sequenced on Hiseq X10 to produce paired-end 150 bp reads. All libraries were summarized in Supplementary Data 1.

**RNA BS-seq.** For _D. mel_, frog, zebrafish, H1, and H9 RNA samples, RNA BS-seq library construction was performed as we previously described[29]. In brief, total RNA was isolated with TRIzol reagent and Direct-zol RNA MiniPrep kit. Poly-adenylated RNA was separated from total RNA using Oligo dT Magnetic Beads (Vazyme). 100-1000 ng of polyadenylated RNA was converted using the EZ RNA methylation kit (Zymo Research) with a modified high-stringency conversion condition. The converted RNA was fragmented into 150–200 nt fragments by incubation at 94 °C for 8 min in fragmentation buffer (NEBNext Ultra II Directional RNA Library Prep Kit, NEB). The fragmented RNA was then used for library construction, following the manufacturer's protocol (NEBNext Ultra II Directional RNA Library Prep Kit, NEB).

For human and mouse oocyte and embryo samples, due to the limited amount of total RNA obtained, RNA BS-seq was performed using rRNA-depleted RNA. rRNA was depleted by Ribominus Eukaryote kit v2 (Thermo, A15020) and concentrated by ethanol precipitation from 100 ng total RNA. Next, the RNA was converted and used for library construction as described above.

Libraries were sequenced on Hiseq X10 to produce paired-end 150 bp reads. All libraries were summarized in Supplementary Data 1.

**Dot blotting.** Dot blotting was performed as we previously described[29]. In brief, the embryos of _D. mel_ W1118 (0.5–1 and 6–8 h) and AB zebrafish (2 and 10 hpf) were collected. Total RNA was isolated, and polyadenylated RNA was further obtained with Oligo dT Magnetic Beads (Vazyme). RNA was then denatured at 95 °C for 3 min, followed by chilling on ice for 5 min. Next, 75 ng RNA was spotted onto a nylon membrane (GE Healthcare) and fixed onto the membrane by cross-linking in a UV Stratalinker at 200mJ. After blocking with 5% BSA in TBST buffer, the membrane was incubated with mouse anti-m5C monoclonal antibody (Diagenode, 15200003, Lot# 003, 1:250) overnight at 4 °C. The membrane was then washed with 1xTBST, followed by incubation with HRP-conjugated goat anti-mouse monoclonal antibody (CST, Cat# 7076 S, Lot# 32) at 1:10,000. Last, the membrane was visualized with Immobilon Western Chemiluminescent HRP Substrate (Millipore). Loading was assessed by methylene blue (Sigma) staining of the membrane.

**NSUN2 staining in human and mouse samples.** Human and mouse oocytes were collected as described above. The collected oocytes were fixed with 4% Paraformaldehyde (PFA) in PBS for 40 min at room temperature. After permeabilization with 1% Triton X-100 for 30 min, oocytes were blocked with 2% BSA for 1 h and incubated with anti-NSUN2 antibody (Proteintech, 20854-1-AP, 1: 200) for 2 h at room temperature. Following the wash with PBS containing 0.3% poly-vinylpyrrolidone, oocytes were incubated with Alexa-Fluor-labeled secondary antibody (Thermo Fisher, A32732, 1:1000) for 1 h at room temperature for the detection of signals.

HeLa cells were plated on 13-mm glass coverslips in six-well plates and grown until cells reached 50–70% confluency. Cells were fixed with 4% PFA in PBS for 15 min, permeabilized in 0.1% Triton X for 30 min, and blocked with 2% BSA for 1 h at room temperature. Next, cells were incubated with anti-NSUN2 antibody (1:200) overnight at 4 °C. Following the wash with PBS, cells were incubated with Alexa-Fluor-labeled secondary antibody (1:1000) for 30 min at room temperature for the detection of signals.

For all staining experiments, DNA was counterstained with DAPI. Leica SP8 confocal microscope was used to obtain the images.

**Hatching assay.** Wild-type or NSUN2 mutant virgins were crossed to wild-type males for 2 days in advance. Hatching rate was determined by counting the number

of 1-day hatched larvae relative to the total number of hatching embryos. The transparent unfertilized eggs were excluded from the analysis.

To compare the development difference between wild-type and mutant embryos, parents were allowed to lay eggs for 10 min and ~200 embryos per condition were collected and incubated at 25 °C. Eighteen hour after oviposition, we counted the number of newborn first larvae every 30 min. The transparent unfertilized eggs were excluded from the analysis.

**DAPI staining and the assessment of *D. mel* embryonic stages.** To visualize the stage of embryos, wild-type or NSUN2 mutant virgins were crossed with wild-type males. Embryos collected on grape juice plates within 2 h were dechorionated by 50% commercial bleach. Dechorionated embryos were fixed by a 1:1 mixture of 4% formaldehyde and heptane for 30 min under rotation. Embryos were then transferred into 1:1 mixture of methanol and heptane and shook vigorously for 30 s. The supernatant was discarded, and the embryos were rinsed in methanol and rotated at 4 °C overnight. After fixation, embryos were sequentially rehydrated in 80%, 50%, and 20% methanol for 10 min each. Embryos were washed for 15 min with PBST three times, stained by DAPI for 30 min at room temperature, and mounted on the slides. The transparent unfertilized eggs were excluded at the beginning of the experiment.

**Cell culture.** HeLa, HEK293T, hESC H1, hESC H9, and 15P-1 cells were purchased from Cell Bank, Type Culture Collection, Chinese Academy of Sciences (CBTCCCAS). These cell lines have been identity verified using short tandem repeat (STR) analysis, which involves the simultaneous amplification of 17 STR markers plus amelogenin to confirm the identity of the cells, by CBTCCCAS. They were also routinely tested for mycoplasma by PCR detection of conditioned medium. NSUN2 knockout HeLa cell line was generated in our previous study[29]. HeLa, HEK293T, and 15P-1 cells were maintained in DMEM (Gibco) supplemented with 10% FBS (HyClone). hESC H1 and H9 were routinely maintained at the undifferentiated state using mTeSR1 (Stem cell 05850), a feeder-free maintenance medium for hESC, following the manufacturer's protocol.

**NSUN2 plasmid construction and transfection.** To construct NSUN2 expression plasmids, total RNAs from zebrafish embryos, mouse 15P-1 cells, and HEK293T cells were reverse transcribed using SuperScript III Reverse Transcriptase (Thermo Fisher), and full-length NSUN2 CDS fragments were amplified. Each NSUN2 CDS fragment was then tagged with 3xFLAG by overlapping PCR and inserted into the EcoRI and BamHI sites of the pCDH vector to generate pCDH-3xFLAG-NSUN2 plasmid. NSUN2 knockout HeLa cells were plated in a 6-well plate and 2 ug plasmids were transfected using Lipofectamine 3000 following the manufacturer's instruction. 48 h after the transfection, cells were collected for RNA isolation and the expression of NSUN2 proteins was confirmed by western blot using anti-FLAG antibody (Sigma-Aldrich, Cat# F1804, Clone# M2, Lot# SLCD6338, 1:1000).

**NSUN2 substrate plasmid construction and transfection.** The ~140 bp substrate pools were synthesized at BGI Shenzhen (Supplementary Data 5). The fragments were amplified, gel purified, and inserted into psiCHECK2 vector by Gibson assembly. The reactions were performed using NEBuilder HiFi DNA Assembly Master Mix (NEB) by mixing the fragments with the linearized vector in a 7:1 molar ratio. Four microliter assembled products were used for bacteria transformation. Bacteria were shaken in SOC medium at 250 rpm for 60 min and then separated on 14 cm LB agar ampicillin selective plates. After 37 °C incubation for about 12 h, eight plates of bacteria were harvested. Plasmids were extracted with the endotoxin-free plasmid extraction kit (TIANGEN).

**Targeted BS-seq.** Total RNA was extracted with TRIzol reagent and Direct-zol RNA Kit 48 h after the transfection. Total RNA was treated with DNase I, BS converted (Sulfonation: 1 cycle, (1) 70 °C, 10 min. (2) 64 °C, 45 min. Desulfonation: 25 °C for 30 min.), and reverse transcribed with gene-specific primers using HiScript II Q RT SuperMix (Vazyme). Target sequences were amplified using STARmix Taq DNA Polymerase (GenStar) with the following program: 94 °C for 3 min; 25 cycles of 94 °C for 30 s, 52 °C for 30 s and 72 °C for 20 s; and 72 °C for 1 min. Next, the PCR product was purified with DNA Clean Beads (Vazyme) and dissolved in 10 μl water. 1 μl purified product was amplified using sequencing adapters with the same program above for five cycles. Finally, the PCR product was recovered with Zymoclean Gel DNA Recovery Kit (Zymo Research) and sequenced. All primers used for targeted BS-seq are listed in Supplementary Data 5.

**Nocodazole treatment of HeLa cells.** HeLa cells were first seeded and grown to 50–70% confluency. The cells were then treated with 0.1 μg/ml nocodazole (Sigma, M1404) for 0, 24, 48, and 72 h. Nocodazole treatment caused prometaphase arrest and apoptosis[54]. In prometaphase, the nuclear membrane broke, and NSUN2 was distributed in the cytoplasm. In addition, global translation was reduced with the induction of apoptosis. Thus, this treatment partially mimicked the status of MI/ MII oocytes, i.e., persistent contact between translationally silenced mRNAs and NSUN2 in the cytoplasm. The expression of NSUN2 protein in different samples

was examined by western blot. The cytoplasmic localization of NSUN2 after nocodazole treatment was confirmed by immunostaining.

**Genome assembly and gene models.** Genome, transcriptome, and gene annotations of *D. mel* BDGP5.78, zebrafish Zv9.78, mouse GRCm38.87, and human GRCh37.75 were downloaded from Ensembl. Xenopus genome assemblies and gene models of *X. laevis* v9.2 and *X. tropicalis* v9.1 were downloaded from Xenbase. Xenopus gene features were extracted from GFF3 files.

**BS-seq data analysis.** mRNA m5C sites were called as we previously described[29]. In brief, we first trimmed adapters, the first 10 bp of the reads, the last 6 bp of the reads, and the low-quality bases using cutadapt[55] and Trimmomatic[56]. Then clean reads were mapped to the in silico converted genome by HISAT2[57] to obtain unique alignments. The remaining unmapped and multiple mapped reads were further mapped to the in silico converted transcriptome by Bowtie2[58]. Alignment results were merged, and only bases with high quality ($Q \geq 30$) were used for the variant calling. To remove noise, a series of filters were applied. In brief, we inferred the mismatch type of each site based on the strand of overlapping annotated genes. We inspected all positions with C-to-T mismatches and only took variant positions into consideration if they conformed to our requirements for the number, frequency, and quality of bases that vary from the converted reference sequences. We specifically required that each variant was supported by three or more variant nucleotides having a base quality score of $\geq 30$, mismatch frequency $\geq 0.1$ and coverage of $C + T \geq 20$. Furthermore, we required that (i) the variant still satisfies the above criteria after the removal of the overlapped C-reads based on the Gini coefficient determined C-cutoff filter, (ii) the signal ratio of the variant is $\geq 0.9$, (iii) the variant is not located at conversion-resistant genes and (iv) the p-value calculated using one-sided binomial test based on gene-specific conversion rate is <0.001. Last, to determine the set of high-confidence sites in a specific sample, we required the presence (mismatch frequency $\geq 0.1$) of a site in both replicates. Sites with a combined p-value (Stouffer's Z-score method) < 0.001 were considered as high-confidence m5C sites. If a site is only present (mismatch frequency $\geq 0.1$) in one of the replicates but not others due to the coverage issue, we further required at least five variant nucleotides in that sample to achieve high specificity.

Based on the Gini coefficients calculated with different C-cutoff filters, we chose C-cutoff = 3 for high-confidence site call. Notably, the observations about the extensive methylation of maternal transcriptome and the limited number of m5C sites at the later embryonic, larval, pupal, and adult stages still hold regardless of which C-cutoff filters we used in all species studied (Supplementary Fig. 3a).

**Motif discovery and m5C site classification.** MEME (v5.0.0) was used for motif discovery[35]. In brief, m5C sites and their 25 nt flanking sequences were extracted and processed to MEME (-rna -objfun ce -nmotifs 5 -cefrac 0.1 -minw 5 -maxw 8) to search for the best five 5–8 nt motifs. Motifs that do not contain the m5C position were not considered. Since MEME and FIMO algorithms tended to introduce both false positive and false negative results, excepted for the analysis in Supplementary Data 3, we classified sites with 5'-CUCNA-3' (m5C underlined) motif as Type II sites and then classified the remaining sites as Type I sites.

Notably, the initial studies from us[31] and other groups[32,33] identified 5'-CUCCA-3' as the motif of NSUN6-dependent sites. However, as we showed in our previous study[31], the alternative bases at position +3 were tolerant by NSUN6. For example, in human adult tissues and adult *D. mel*, 7.5% and 39.7% of bases at position +3 in Type II sites were not cytosines, respectively. We have further experimentally verified this observation by NSUN6 knockout experiment in *D. mel* and human Type II substrate mutagenesis experiment[31]. Thus, to identify Type II sites more accurately in multiple species we analyzed, we used an extended Type II motif, 5'-CUCNA-3', in this study. We also confirmed that 5'-CUCNA-3' motif was not favored by NSUN2 by human NSUN2 substrate mutagenesis assay in this study. Thus, this extension is unlikely to introduce Type I to Type II misclassification.

For motif generation, the m5C sites and flanking regions were extracted from the transcriptome. Motif logos were plotted with WebLogo 3.5[59].

**m5C density calculation.** m5C density was defined as the ratio of m5C number to the total C number in the background. An overall m5C density in a sample was calculated by dividing the number of m5C sites by the total number of Cs passed the coverage filter (≥20 reads) in BS-seq. To describe the m5C densities along transcripts, we grouped the transcripts into bins and the m5C density in each bin was calculated by dividing the m5C number in a bin by the total C number in the corresponding bin. Because the average transcript lengths vary among species, to make it comparable, we fixed 5'UTRs as ten bins (or five bins for analyses in Figs. 3b and 4b) and computed the bin numbers of CDS and 3'UTR by their relative average lengths to the corresponding 5'UTR. m5Cs and background Cs were then grouped into bins to calculate m5C densities.

**m5C site conservation analysis.** LiftOver tool was used to convert the genome position among human, mouse, zebrafish, and two frog species. Chain files (hg19, mm10, danRer7, xenTro3, xenTro9 and xenLae2) downloaded from UCSC were used.

Conserved methylation sites were defined as m5C sites that were with a level ≥10% in one species and with a level >5% in another species. To determine whether a site is unmethylated, we required that the site is covered by ≥20 reads and the level is <5%. Only sites that can be defined as methylated or unmethylated were analyzed.

**RNA-seq analysis**. Adapters were removed from paired-end reads with cutadapt[55] (-q 25 -e 0.1). Clean reads were mapped to the reference genome and transcriptome with Tophat v2.1.1[60] in strand-specific manner (--library-type fr-firststrand). Unique alignments were processed to Cufflinks v2.2.1[61] for FPKM calculation. Average FPKM values of replicates were used.

**Principal component analysis**. PCA was carried out using scikit-learn (https://github.com/scikit-learn/scikit-learn). Thirteen thousand six hundred twenty-six genes with FPKM value > 0 in at least one sample were selected. Log2 transformed data were projected to 6 dimensions. The first three components together captured 91.3% of the variance.

**RNA structure prediction and comparison**. To predict the secondary structure, the upstream and downstream 50 nt sequences of the m5C sites were extracted from the transcriptome and folded with the RNAfold tool in ViennaRNA Package[62]. Since some of the species used were ectothermic animals, the folding temperatures were adjusted to their culturing temperatures: 25 °C for *D. mel* and *X. tropicalis*, 28.5 °C for zebrafish, and 22 °C for *X. laevis*. For humans and mice, 37 °C were used.

To compare the difference of base-pairing frequencies in each position between two selected samples, we used the bootstrapping method to calculate the p-values. In brief, we first down-sampled 100 sequences for each sample and calculated the frequencies of base-pairing in each position. This process was repeated 1000 times. Next, we tested the difference between two samples using one-sided Student's t-test.

**Gene ontology (GO) analysis**. GO analysis was performed with R package ClusterProfiler (v3.10.1). Human (org.Hs.eg.db) gene annotation using Entrez Gene identifiers was used in the enrichment calculation of Biological Process. Benjamini and Hochberg's FDR correction was used in P-values adjustment.

**Logistic GLM for m5C sequence and structural feature analysis**. To evaluate the contribution of motif sequence and structure to evolutionary changes of m5C between species, we performed logistic GLM analysis using the non-conserved sites in human-mouse or *X. tropicalis-X. laevis* pair. Using human-mouse pair as an example, human or mouse m5C sites were first mapped to mouse or human genome by liftOver, and the ones that were Cs at the DNA level in both species but only methylated in one species were retained. Next, sequences in the species with methylation detected were used as positive sets, while sequences in the species with no methylation were used as negative sets. Then the following features were extracted: (1) for Type I sites, the base composition of positions −2 to +5 (m5C position was omitted because bases in this position were all C) and RNA structures of the 50 nt flanking sequence; (2) for Type II sites, RNA structures of positions −20 to +20 were used. These features were dummied into binaries (for structures, paired status was considered as 1) and processed to glm function (exponential family distribution with logit transformation) in Python package statsmodels (v0.10.1). Feature-feature interactions were not considered in this analysis.

**Targeted BS-seq analysis**. Reads were mapped to the reference sequences (DNA library against the original reference sequences; BS-seq library against the C-to-T converted reference sequences) with Bowtie2 (--norc). Custom scripts were used to extract reads with barcodes that were unique in both DNA and BS-seq libraries. The C/T counts at m5C position were extracted and the methylation level was defined as the number of reads with C divided by the number of reads with C or T.

**Reporting summary**. Further information on research design is available in the Nature Research Reporting Summary linked to this article.

## Data availability

The non-human data generated in this study have been deposited in the NCBI's Gene Expression Omnibus database under accession code GSE127780 and GSE127781. The human data generated in this study have been deposited into CNGB Sequence Archive of China National GeneBank under accession code CNP0001844. Source data are provided with this paper.

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

## Acknowledgements

We thank Jin Billy Li and Nannan Gu for the discussion of the manuscript. We thank Dr. Stephan Sigrist for sharing the *D. mel* UAS-dNSun2 stain, Dr. Chung-I Wu for sharing the equipment and space for *D. mel* experiments, Dr. Xiaohang Yang for sharing the *D. mel* CRISPR/Cas9 tool stains, Dr. Xionglei He for sharing the zebrafish strain, Dr. Yonglong Chen for sharing *Xenopus tropicalis* samples, Xenopus Resource Center for the collection of *Xenopus laevis* embryos, and SYSU Ecology and Evolutionary Biology Sequencing Core Facility for the sequencing service. We thank Shuaiqi Zhao for the assistance of frog embryo collection. This study was supported by grants from National Key R&D Program of China (2018YFC1003100 to R.Z. and Y.X.; 2018YFC1003200 to W.G.), Guangdong Major Science and Technology Projects (2017B020226002 to R.Z.), Guangdong Innovative and Entrepreneurial Research Team Program (2016ZT06S638 to R.Z.), National Natural Science Foundation of China (31571341 and 91631108 to R.Z.; 81802826 and 82173050 to T.H.; 31970668 to W.G.), Guangdong Natural Science Foundation (2021A1515010667 to T.H.), and Postdoctoral Science Foundation of China (2019M663223 to T.H.).

## Author contributions

R.Z., Y.X., C.Z., and W.Z.G. contributed to the study design. T.H., W.C., X.N.Z., and M.X. performed all experiments in *D. mel*. C.D., B.C., T.Z., Y.Z., and S.L. performed human/mouse oocyte/embryo sample collection and BS-seq. T.Z. performed zebrafish/frog sample collection and BS-seq. X.H performed human ES cell culture experiment. Y.Z. performed high-throughput mutagenesis assay and NSUN2 OE experiment. L.Z. and W.C. performed NSUN2 immunostaining experiment. J.L. performed bioinformatics analysis. R.Z. and J.L. wrote the manuscript with input from all authors.

## Competing interests

The authors declare no competing interests.
