## [Peer Review File · Nature Communications]

Title: Developmental mRNA m5C landscape and regulatory innovations of massive m5C modification of maternal mRNAs in animalsREVIEWER COMMENTS

Reviewer #1 (Remarks to the Author):

In this manuscript, the authors surveyed the developmental dynamics of m5C modification in mRNAs in an unprecedented resolution and scale in six animal species, including from *Drosophila* to human embryos. Their analysis reveals unexpectedly high levels of methylated cytidines in maternal RNAs, and they are dramatically remodeled during the developmental progression. Most identified sites fall into one of two types, which harbor the characteristic sequence motif and secondary structure of NSUN2 and NSUN6, respectively. NSUN2 looked highly efficient at methylation of maternal mRNAs in a meiotic phase despite its relatively low mRNA expression. The authors could successfully show that prolonged exposure of NSUN2 in the cytoplasm is enough for such high levels of mRNA methylation using cell cycle-arrested HeLa. Also, they analyzed distributions of binding sites, functional and developmental consequences, and evolutionary conservation of the NSUN2/NSUN6-driven cytosine methylation in the early maternal-to-zygotic transition in animal development.

This study delivers tons of valuable new observations with thorough analyses that improve our understanding of how the messenger RNAs are controlled in the animal MZT. Additional experiments and analyses addressed many possible questions about the phenomenon.

There is no doubt that the publication of this manuscript is beneficial to the scientific community. Still, I find some points to be addressed for a more precise interpretation of the discoveries.

- I see insufficient support for the notion that NSUN6 is responsible for most type II sites. Although sequence similarity of substrates does not always come from a single protein, the manuscript does not differentiate between type II sites and NSUN6 targets enough throughout the text.

- Line 132-136: The enrichment in 5' UTR and type I/II difference in distribution are a lot less clear in Fig 1d. Despite the description "Type II sites were distributed evenly along transcripts," the curves for type I and II seem to be far less distinctive if they are scaled to show just the site distribution regardless of the absolute methylation level.

- Line 192: Are NSUN2-null males fertile? It should be helpful to clearly distinguish whether the developmental delay of NSUN2 is from maternal deposits or zygotic transcripts if NSUN2 mutant father is included in this experiment.

- Fig 4c: I am not perfectly persuaded by what we are expected to learn from this panel. It should feel convincing if this is backed by statistical analysis for experimental variability and significance with replicates of different BS-seq libraries.

- Line 273–274: There should be several alternative explanations about the failure of mimicking the MII oocytes with HeLa other than "the existence of oocyte-specific regulation."

- Line 274-276: Frogs and zebrafish have significantly higher levels of NSUN6 transcript than humans in the early stages. They also have similar dynamics to that of humans. Do they share a majority of target mRNA? Do NSUN6 in frogs and zebrafish also target mRNAs related to mitotic cell cycle regulation?

- Line 289-290: How about the number of methylation sites or overall methylation rate of each mRNA or each 5' UTR? Are they conserved?

Reviewer #2 (Remarks to the Author):

This is an interesting and well done manuscript that seeks to characterize m5C a previously controversial nucleotide modification and mRNA. This group had a very important paper in 2019 in NSMB that provided the first believable map of m5C in the transcriptome. Now this group uses their mapping technologies and demonstrate that there is a sizable and unexpected increase in m5C during specific stages of embryo development. The authors use flies predominantly, but also other tissues to demonstrate the general phenomenon of this increase in methylation. The authors link this to nuclear membrane disruption and increased access of the mRNAs to the modifying enzymes. They do a good job identifying and sent to as a particularly important enzyme in this process of methylation.

Overall, I thought the experiments were compelling, and evidence for the specific enzymatic pathway were fairly solid.

The major concern is the following: readers are going to want to know the exact stoichiometry of m5C sites in normal tissues as well as during this stage is where the methylation goes up dramatically. They want to get numbers: They want to know the exact percent of methylation. Although the authors might be able to extrapolate this using their bisulfite method and standards, it is important to realize that the results described here are surprising and unexpected and therefore an orthogonal method would be particularly important to validate, at least for a handful of m5C sites, the massive increase in methylation that occurs in specific stages. Methods such as SCARLET allow researchers to use a biochemical method to directly determine the exact stoichiometry of any given nucleotide in any mRNA. This type of method could be very powerful and I think would be very important to give greater confidence that the increase in methylation is not due to some weird contaminant that is somehow influencing the bisulfite reactivity, but due to true increases in methylation. Additionally, the high quantitative accuracy of SCARLET can give readers a clear understanding of the true fold-increase in methylation. The use of these orthogonal methods would be very critical to convince in an unambiguous way that the methylation levels are changing.

Reviewer #3 (Remarks to the Author):

The manuscript by Liu et al. provides remarkable new insights into the occurrence, enzymology and

plausible biological function of m5C in mRNAs.

The authors give a to-the point introduction to the background of this field, to which they previously contributed important high-quality data and analyses (ref 26). I have some suggestions here:

1. line 47-48: I suggest to acknowledge that evolutionary aspects of many types of RNA modifications have in fact long been studied, e.g. the distribution of most of them, including their modifying enzymes, across kingdoms of life is known and has often been summarised in reviews and the RNA modification databases.
2. Line 61: An additional reference confirming work in ref 26 could be cited:
<https://doi.org/10.1186/s12915-020-00769-5>
3. The Introduction seems run directly into Results, without the typical final intro para and 'Results' subheading. This could easily be fixed.

The results presented consist of a wealth of high-quality sequencing and diverse other data, which are presented in excellent figures and generally described very well in the figure legends and main text. I concur with all the main conclusions drawn, which are also clearly presented and contribute several important new insights. The criteria for and implementation of the bioinformatic bsRNA-seq analyses derive from ref 26 and are an important aspect of making the results on m5C candidate sites dependable. I have some suggestions for improvement:

1. Implementation of the $\geq 10\%$ minimal cytosine non-conversion threshold is clearly a key aspect of determining m5C candidate site numbers. However, I am not sure how to understand the following statement in line 783: 'we required the presence of a site in both replicates'. Does this mean that the site had to be $\geq 10\%$ in both replicates, or just 'present'? More importantly, with some of the figures I am not sure how sites were included/excluded, especially for preparation of the box plots for 'm5C level'. In Fig 1B, many of the green box plots extend well below the 10% threshold, so some sort of union set of sites that had $\geq 10\%$ in at least one sample must have been used. It could be as described for Fig 2D, please clarify. If so, I am still not quite sure if this is the best way of presenting the data. Clearly, many sites disappear (probably more accurately drop below the $\geq 10\%$ threshold) at later timepoints. So the green box plots should probably show the m5C level only for the sites that remain?! In Fig 2F, I am not sure why there are two distinct black lines for wild type. Is this because sites were separated into type I and II? Please explain better in the legend. For EDFig 2 the issue is clearly explained in the legend but this information is again not stated in EDFig 12A. Please add to legend.
2. Lines 217-219: I'm not sure how exactly the prior YBX1 findings complement the present work but this is probably best explained in the discussion. More importantly, I don't follow the conclusion that the work ESTABLISHED the functional significance of mRNA m5C. The problem is of course that NSUN2 also methylated tRNAs (acknowledged in the discussion). I would say 'suggests' or 'indicates' rather than 'establish'.
3. Several figure panels show graphs that lack error bars. These should be added or else it should be made clear that this cannot be done either due to lack of replication or because of the nature of the data treatment.
4. Visual/qualitative data in gels, blots and microscopy images should also be accompanied with statements in the legends that indicate the findings were equivalent in a number of replicate

experiments.

The discussion is again succinct and clear. Again, I have some suggestions:

1. Lines 338-342: The way in which prior work with YBX1 is supportive of a function of m5C in maternal mRNA should be made more explicit. This is critical, as consequences of NSUN2 loss alone do not show this given that it is primarily known as a tRNA modifying enzyme.
2. Lines 347-349: I am not convinced that the study has shown that higher structure in 5' regions LED to gain of NSUN2-mediated m5C sites. The data is certainly consistent with that and suggests that but is it proof of causality?
3. Lines 349-352: This statement should be backed up by citing the literature that indicated a link between m5C and translation. At least one of these studies (<https://doi.org/10.1186/s12915-020-00769-5>) narrows this link to mRNAs where the m5C sites are around the start codons.

Supplementary discussions. These make important points and they are quite short and I wonder whether they cannot simply be made part of the main manuscript.

We thank the reviewers for examining our manuscript and providing constructive feedback. We are very grateful that all reviewers agreed on the merit of our study. We performed additional analyses and experiments to address questions raised by the reviewers. The changes are highlighted in **Yellow** in the revised manuscript for easy tracking. Please see below for our point-by-point response to reviewers' comments.

Reviewer #1 (Remarks to the Author):

In this manuscript, the authors surveyed the developmental dynamics of m5C modification in mRNAs in an unprecedented resolution and scale in six animal species, including from *Drosophila* to human embryos. Their analysis reveals unexpectedly high levels of methylated cytidines in maternal RNAs, and they are dramatically remodeled during the developmental progression. Most identified sites fall into one of two types, which harbor the characteristic sequence motif and secondary structure of NSUN2 and NSUN6, respectively. NSUN2 looked highly efficient at methylation of maternal mRNAs in a meiotic phase despite its relatively low mRNA expression. The authors could successfully show that prolonged exposure of NSUN2 in the cytoplasm is enough for such high levels of mRNA methylation using cell cycle-arrested HeLa. Also, they analyzed distributions of binding sites, functional and developmental consequences, and evolutionary conservation of the NSUN2/NSUN6-driven cytosine methylation in the early maternal-to-zygotic transition in animal development.

This study delivers tons of valuable new observations with thorough analyses that improve our understanding of how the messenger RNAs are controlled in the animal MZT. Additional experiments and analyses addressed many possible questions about the phenomenon.

There is no doubt that the publication of this manuscript is beneficial to the scientific community. Still, I find some points to be addressed for a more precise interpretation of the discoveries.

- I see insufficient support for the notion that NSUN6 is responsible for most type II sites. Although sequence similarity of substrates does not always come from a single protein, the manuscript does not differentiate between type II sites and NSUN6 targets enough throughout the text.

We thank the reviewer for this comment. We apologize for not clearly describing the rationale behind the determination of NSUN6 as the writer protein of Type II sites. We previously showed that NSUN6 is responsible for sites containing a downstream TCCA motif (i.e. Type II sites) in human HEK293T and HeLa cells¹. Thus, in this study, we assume that all sites with this motif were Type II sites and methylated by NSUN6. We agree with the reviewer that ideally, direct evidence needs to be provided to demonstrate that NSUN6 targets all Type II sites in maternal RNAs. To confirm this, we collected 0 h embryos from NSUN6 knockout flies we generated previously¹ and performed BS-seq. Of the 348 Type II sites identified in wild-type flies, the majority of them lost methylation in knockout embryos. This observation suggests that NSUN6 may methylate the majority of Type II sites in fly oocytes. In the revised manuscript, we also point out although the sequence and structural

features suggest that Type II sites in maternal mRNAs were likely NSUN6 targets, future studies based on knockout or knockdown models are needed to confirm NSUN6 as the writer protein of Type II sites in maternal mRNAs in vertebrates (**Page 14, Lines 388-392**).

Fig. R1. Comparison of m5C methylation levels in the 0 h embryos between wild-type flies and NSUN6 knockout flies. Sites identified in the wild-type sample were analyzed.

- Line 132-136: The enrichment in 5' UTR and type I/II difference in distribution are a lot less clear in Fig 1d. Despite the description "Type II sites were distributed evenly along transcripts," the curves for type I and II seem to be far less distinctive if they are scaled to show just the site distribution regardless of the absolute methylation level.

We thank the reviewer for this comment. To address this question, we have added a figure to show the normalized distribution of Type I and Type II sites (**Extended Data Fig. 5e**). We agree with the reviewer that Type I and Type II sites had a similar distribution among CDS and 3'UTR regions. We also found that the conclusion about the 5' end enrichment of Type I sites in mammals is still held. We have revised the sentence to clarify this (**Page 6, Lines 150-152**).

“In all species, Type I and Type II sites had a similar distribution in CDS and 3'UTR regions (**Fig. 1d, Extended Data Fig. 5e**).”

Extended Data Fig. 5e. The normalized distribution of maternal m5C sites along the transcripts in different species. In this analysis, to have a fair comparison of the Type I and Type II site distribution, the individual distributions in Fig. 1d were standardized with their maximum value as 1 separately.

- Line 192: Are NSUN2-null males fertile? It should be helpful to clearly distinguish whether the developmental delay of NSUN2 is from maternal deposits or zygotic transcripts if NSUN2 mutant father is included in this experiment.

In all our experiments, we crossed female NSUN2 mutant flies with male wild-type flies to produce maternal NSUN2 knockout embryos, thus the developmental delay should be from maternal transcripts. We have clarified this in the result section.

- Fig 4c: I am not perfectly persuaded by what we are expected to learn from this panel. It should feel convincing if this is backed by statistical analysis for experimental variability and significance with replicates of different BS-seq libraries.

We thank the reviewer for this suggestion. To estimate the statistical significance of base-pairing frequency difference of Type II m5C sites and flanking regions between human MII oocytes and other samples (hESC H1, human TE, and mouse MII oocytes). We used the bootstrapping method to calculate the p values for each position. In brief, we first down-sampled 100 sequences for each sample and calculated the frequencies of base-pairing in each position. This process was repeated 1000 times. Next, we tested the difference between two samples using one-sided student's t-test. We found that, in most positions, human MII sites had significantly lower unpairing frequencies (loop region) and lower pairing frequencies (stem region) than other samples (**Extended Data Fig. 11d**), consistent with the metaprofile results.

Extended Data Fig. 11d. Comparison of the structural difference of Type II m5C sites and flanking regions between human MII oocytes and other samples (hESC H1, human TE, and mouse MII oocytes). The bootstrapping method was used and p values were calculated using one-sided student's t-test (Methods). Positions in human MII oocytes with significantly lower unpairing frequencies (loop region) and lower pairing frequencies (stem region) than other samples are indicated with up and down arrows, respectively.

- Line 273–274: There should be several alternative explanations about the failure of mimicking the MII oocytes with HeLa other than "the existence of oocyte-specific regulation."

We thank the reviewer for pointing this out. We have listed some possible mechanisms we can think of (**Page 11, Lines 289-293**), and we hope that future studies based on CRISPR knockout or activation screening may help to reveal the

mechanism.

“..., This result might be explained by the oocyte-specific regulation of NSUN6, the existence of oocyte-specific factors, such as RNA binding proteins, to cooperate with NSUN6 to promote methylation, or the presence of a negative regulator that represses Type II site methylation in HeLa cells.”.

- Line 274-276: Frogs and zebrafish have significantly higher levels of NSUN6 transcript than humans in the early stages. They also have similar dynamics to that of humans. Do they share a majority of target mRNA? Do NSUN6 in frogs and zebrafish also target mRNAs related to mitotic cell cycle regulation?

Although frogs and zebrafish have significantly higher levels of NSUN6, their type II sites are much less than human sites. In zebrafish 0 hpf embryos, there were only 135 genes methylated by NSUN6 (**Fig. R2a**) and no Biological Process terms were significantly enriched. In *X. laevis*, there were 1355 genes with Type II sites. These genes were only slightly significantly enriched in Biological Process terms related to cytoskeleton, metabolism, and protein modifications (**Fig. R2b**).

Fig. R2. Frogs and zebrafish Type II site analysis.

(a) Overlaps of the genes with Type II sites between humans, frogs, and zebrafish.

(b) Enrichment map plot (see Methods) for genes in *X. laevis* stage 0 sample. *X. tropicalis* sample was not analyzed because no suitable gene annotation database was available.

- Line 289-290: How about the number of methylation sites or overall methylation

rate of each mRNA or each 5' UTR? Are they conserved?

At the gene level, we only observed weak cross-species conservation of methylation site numbers, as exemplified in human-mouse pair and *X. laevis*-*X. tropicalis* pair (Extended Data Fig. 12b). Moreover, the overall methylation levels of individual genes were not conserved between species (Extended Data Fig. 12c). We have added these results in the revised manuscript (Page 12, Lines 310-314).

Extended Data Fig. 12b&12c. Scatterplot showing the number of m5C sites or median m5C levels of orthologous genes or 5'UTR of orthologous genes in human-mouse pair and *X. laevis* -*X. tropicalis* pair. For humans and mice, methylation sites or levels obtained from MII oocytes were used. For frog species, methylation sites or levels obtained from stage 0 embryos were used. Sites that are covered by ≥ 20 reads in both species were analyzed.

Reviewer #2 (Remarks to the Author):

This is an interesting and well done manuscript that seeks to characterize m5C a previously controversial nucleotide modification and mRNA. This group had a very important paper in 2019 in NSMB that provided the first believable map of m5C in the transcriptome. Now this group uses their mapping technologies and demonstrate that there is a sizable and unexpected increase in m5C during specific stages of embryo development. The authors use flies predominantly, but also other tissues to demonstrate the general phenomenon of this increase in methylation. The authors link this to nuclear membrane disruption and increased access of the mRNAs to the modifying enzymes. They do a good job identifying and sent to as a particularly important enzyme in this process of methylation.

Overall, I thought the experiments were compelling, and evidence for the specific enzymatic pathway were fairly solid.

The major concern is the following: readers are going to want to know the exact stoichiometry of m5C sites in normal tissues as well as during this stage is where the methylation goes up dramatically. They want to get numbers: They want to know the exact percent of methylation. Although the authors might be able to extrapolate this using their bisulfite method and standards, it is important to realize that the results described here are surprising and unexpected and therefore an orthogonal method

would be particularly important to validate, at least for a handful of m5C sites, the massive increase in methylation that occurs in specific stages. Methods such as SCARLET allow researchers to use a biochemical method to directly determine the exact stoichiometry of any given nucleotide in any mRNA. This type of method could be very powerful and I think would be very important to give greater confidence that the increase in methylation is not due to some weird contaminant that is somehow influencing the bisulfite reactivity, but due to true increases in methylation. Additionally, the high quantitative accuracy of SCARLET can give readers a clear understanding of the true fold-increase in methylation. The use of these orthogonal methods would be very critical to convince in an unambiguous way that the methylation levels are changing.

We thank the reviewer for the suggestion. We understand that cross-validation of the methylation sites is important, and we agree that SCARLET is a method to quantify RNA modification levels with high accuracy at a single-base resolution. However, since our lab has no prior experience to perform radioactive material experiments, it will take a long time for us to obtain the permit for conducting radioactive material experiments and complete radiation safety training according to the policy of our university. Moreover, because SCARLET needs a relatively large amount of mRNA for the experiment (e.g. 1ug per replicate), it is difficult to collect enough oocyte mRNAs for the experiment. Thus we are sorry that we are unable to conduct SCARLET accordingly. To address the reviewer's concern, we tried two additional methods for single-base m5C detection in fly samples: (1) SELECT (a single-base elongation and ligation-based qPCR amplification method); (2) Nanopore direct RNA sequencing.

1. SELECT experiment. SELECT was initially developed for m6A detection, but not verified for m5C. To test whether the method also works for m5C, we applied SELECT on a synthesized m5C-containing oligo. Unfortunately, no difference was observed between the m5C-containing oligo and the non-m5C control oligo (**Fig. R3a**). As a comparison, we successfully used SELECT to detect the well-known MALAT1 position 2515 m6A site following the original design² (**Fig. R3b**). Hence, SELECT is currently not suitable for m5C detection.

2. Nanopore direct RNA sequencing. We have an ongoing project to develop a deep-learning method to call m5C sites using nanopore direct RNA-sequencing, and we have generated nanopore data for *D.mel* 0 h embryos and a fully matched modification-free transcriptome that was made by in vitro transcription (IVT). A recent study found that m5C modified RNA sequences presented decreased quality scores, higher mismatch frequencies, higher insertion frequency, particularly in the neighboring residues (position -1 and +1) of m5C sites³. We thus examined these features in our samples to verify whether the sites we identified in maternal mRNAs via BS-seq were likely real. First, we examined in vitro generated m5C modified RNA and un-modified RNAs and confirmed the frequent mismatches in m5C modified RNA (**Fig. R4a**). Next, we compared 0 h embryo sample with IVT sample. We found that the sites called from BS-seq had decreased quality scores, higher mismatch frequencies, higher insertion frequency in the wild-type sample as compared to the modification-free transcriptome (**Fig. R4b-c**), confirming that these sites are enriched in real m5C sites. Note that these features are not robust and quantitative enough for m5C quantification. We are in progress to develop an

algorithm to estimate the per-site modification stoichiometries with nanopore data as the recent attempt on Nm and Ψ RNA modifications³ and plan to prepare a separate manuscript soon.

In summary, we apologize that we could not fully address the reviewer's concern based on our experiment and analysis. We hope that the nanopore data above may provide additional support for the validity of maternal m5C sites identified in this study.

Fig. R3. The attempt of the SELECT method in RNA m5C detection.

(a) qPCR Ct values for a synthesized m5C site. RNA mixtures with known m5C fractions obtained by mixing m5C-Oligo with non-m5C-Oligo were subjected to SELECT analysis. m5C-Oligo was mixed with non-m5C-Oligo at different ratios (0%, 25%, 50%, 75% and 100%).

(b) Normalized abundance of m6A at MALAT1 m6A2515 detected by SELECT. The MALAT1 position m6A2515 site and the flanking position A2511 site that were previously validated using the SELECT method were selected². The experiment was performed as previously described². The Mettl3 knockout cells were generated from our previous study⁴.

Fig. R4. Nanopore RNA sequencing based m5C verification.

(a) IGV snapshots illustrating the differences in mapping for unmodified and m5C containing RNA when base-called with GU 3.0.3. Positions with mismatch

frequencies greater than 0.1 are colored; gray represents match to reference.
(b) Comparison of base-calling features (base quality, mismatch, deletion and insertion frequency) of m5C sites between 0 h embryo sample and IVT sample. WT, 0 h embryo sample; IVT, a modification-free transcriptome. To generate IVT sample, we first generated cDNA with a T7 promoter sequence using RNA from 0 h embryo sample. The cDNA was in vitro transcribed back to RNA using T7 RNA polymerase.
(c) Representative IGV snapshots of 3 m5C sites identified in 0 h embryo sample. m5C sites are indicated above each snapshot by red arrows.

Reviewer #3 (Remarks to the Author):

The manuscript by Liu et al. provides remarkable new insights into the occurrence, enzymology and plausible biological function of m5C in mRNAs.

The authors give a to-the point introduction to the background of this field, to which they previously contributed important high-quality data and analyses (ref 26). I have some suggestions here:

1. line 47-48: I suggest to acknowledge that evolutionary aspects of many types of RNA modifications have in fact long been studied, e.g. the distribution of most of them, including their modifying enzymes, across kingdoms of life is known and has often been summarised in reviews and the RNA modification databases.

We thank the reviewer for this suggestion. We apologize for not clearly describing the current understanding about the conservation and evolution of RNA modification enzymes and sites. We have revised the text to point it out (**Page 3, Lines 46-53**).

“The conservation and evolution of RNA modification enzymes have been extensively studied (e.g.6, 7). Moreover, the distribution of RNA modification sites in noncoding RNA species, such as tRNA and rRNAs, have been described in different species^{1, 8, 9}. Recent technical advances have also revealed a number of mRNA modifications. However, except for m6A methylation and adenosine-to-inosine RNA editing⁹⁻¹², the in vivo functions and evolution of most types of mRNA modifications have not been fully investigated.”.

2. Line 61: An additional reference confirming work in ref 26 could be cited:
<https://doi.org/10.1186/s12915-020-00769-5>

Thanks. We have included this reference in the revised manuscript (**Page 3, Line 69**).

3. The Introduction seems run directly into Results, without the typical final intro para and ‘Results’ subheading. This could easily be fixed.

We thank the reviewer for this reminder. We have added the “Results” subheading and stated the objectives and major findings of the work at the end of the introduction section, as the reviewer suggested (**Page 4, Lines 73-81**).

“To understand the landscape, function, and evolution of mRNA m5C, we sequenced

samples from 6 animal species spanning 800 million years of evolution to construct quantitative maps of mRNA m5C at different stages of development. Unexpectedly, we observed mRNA m5C as a specialized modification that is largely restricted to maternal mRNAs. We further used cell models and animal models to investigate the mechanism underlying the extensive methylation of maternal mRNAs and the biological importance of m5C in early embryonic development. Finally, we applied comparative epitranscriptomic approaches to reveal two major m5C regulatory innovation steps and the rapid evolution of individual m5C sites.”.

The results presented consist of a wealth of high-quality sequencing and diverse other data, which are presented in excellent figures and generally described very well in the figure legends and main text. I concur with all the main conclusions drawn, which are also clearly presented and contribute several important new insights. The criteria for and implementation of the bioinformatic bsRNA-seq analyses derive from ref 26 and are an important aspect of making the results on m5C candidate sites dependable. I have some suggestions for improvement:

1. Implementation of the $\geq 10\%$ minimal cytosine non-conversion threshold is clearly a key aspect of determining m5C candidate site numbers. However, I am not sure how to understand the following statement in line 783: ‘we required the presence of a site in both replicates’. Does this mean that the site had to be $\geq 10\%$ in both replicates, or just ‘present’?

We meant that the sites had to be $\geq 10\%$ in both replicates. We have clarified this in the revised manuscript (**Page 35, Line 817**).

More importantly, with some of the figures I am not sure how sites were included/excluded, especially for preparation of the box plots for ‘m5C level’. In Fig 1B, many of the green box plots extend well below the 10% threshold, so some sort of union set of sites that had $\geq 10\%$ in at least one sample must have been used. It could be as described for Fig 2D, please clarify. If so, I am still not quite sure if this is the best way of presenting the data. Clearly, many sites disappear (probably more accurately drop below the $\geq 10\%$ threshold) at later timepoints. So the green box plots should probably show the m5C level only for the sites that remain?!

We thank the reviewer for this comment. we have revised Fig. 1b to show the sites that remain as the reviewer suggested.

For Fig. 2d, we used a union set of sites with levels $\geq 10\%$ in at least one sample, since the purpose of this analysis is to show how m5C changed during nocodazole treatment in HeLa cells. We have clarified this in the revised figure legend (**Page 19, Lines 437-438**).

In Fig 2F, I am not sure why there are two distinct black lines for wild type. Is this because sites were separated into type I and II? Please explain better in the legend. For EDFig 2 the issue is clearly explained in the legend but this information is again not stated in EDFig 12A. Please add to legend.

Yes, in Fig. 2f, Type I and II sites were sorted and plotted, respectively. We have modified Fig. 2f to point this out.

In Extended Data Figure 12a, we have labeled Type I and II sites as the reviewer suggested.

2. Lines 217-219: I'm not sure how exactly the prior YBX1 findings complement the present work but this is probably best explained in the discussion. More importantly, I don't follow the conclusion that the work ESTABLISHED the functional significance of mRNA m5C. The problem is of course that NSUN2 also methylated tRNAs (acknowledged in the discussion). I would say 'suggests' or 'indicates' rather than 'establish'.

We thank the reviewer for this suggestion. We have rephrased the conclusion accordingly (**Page 9, Line 234**).

“... suggest the functional significance of maternal mRNA m5C”).

We also explained why the two works may complement each other to suggest the functional significance of maternal mRNA m5C in the discussion section (**Pages 13-14, Lines 361-372; please see our response to the discussion part Q1 for details**).

3. Several figure panels show graphs that lack error bars. These should be added or else it should be made clear that this cannot be done either due to lack of replication or because of the nature of the data treatment.

We thank the reviewer for this suggestion. For some of the figure panels, there are no error bars because of the nature of the data treatment. For the ones with replicates, we have added confident intervals or error bars accordingly (**Extended Data Fig. 6, Extended Data Fig. 10a**).

4. Visual/qualitative data in gels, blots and microscopy images should also be accompanied with statements in the legends that indicate the findings were equivalent in a number of replicate experiments.

We thank the reviewer for this suggestion. We have indicated that the findings were consistent in replicates in the revised figure legends (**Fig. 2a, Extended Data Fig. 3b, Extended Data Fig. 7b**).

The discussion is again succinct and clear. Again, I have some suggestions:

1. Lines 338-342: The way in which prior work with YBX1 is supportive of a function of m5C in maternal mRNA should be made more explicit. This is critical, as consequences of NSUN2 loss alone do not show this given that it is primarily known as a tRNA modifying enzyme.

We thank the reviewer for this suggestion. We have stated the limitation of our study and explained why the prior work on mRNA m5C reader protein YBX1 is supportive of a function of m5C in maternal mRNA (**Pages 13-14, Lines 361-372**).

“Notably, a limitation of our study is that we were unable to determine if the observed effects of NSUN2 loss of function on embryogenesis are due to changes in mRNAs or

tRNAs. A previous study in zebrafish found that YBX1 preferentially bound m5C-containing mRNAs and YBX1 interference led to early gastrulation defects, suggesting that maternal mRNA m5C might play a role in early embryonic development³⁹. However, because YBX1 has been linked to a wide range of processes that are unrelated to m5C, the data that directly reflect the functional significance of mRNA m5C and methyltransferase is still missing. As our study identified the major methyltransferase responsible for the maternal mRNA m5C methylation in *D. mel* and observed a similar phenotype upon its knockout, the previous YBX1 study and our study complement each other to suggest the functional significance of maternal mRNA m5C.”

2. Lines 347-349: I am not convinced that the study has shown that higher structure in 5' regions LED to gain of NSUN2-mediated m5C sites. The data is certainly consistent with that and suggests that but is it proof of causality?

We thank the reviewer for this comment. We agree with the reviewer that they were consistent but not proof of causality. We have modified the statement as “the emergence of more structured 5' end regions in mammals was accompanied with the gain of NSUN2-mediated m5C sites at the 5' end of the mRNAs.”

3. Lines 349-352: This statement should be backed up by citing the literature that indicated a link between m5C and translation. At least one of these studies (<https://doi.org/10.1186/s12915-020-00769-5>) narrows this link to mRNAs where the m5C sites are around the start codons.

Thanks. We have cited the reference as the reviewer suggested (**Page 14, Line 382**).

Supplementary discussions. These make important points and they are quite short and I wonder whether they cannot simply be made part of the main manuscript.

We thank the reviewer for this suggestion. We have included Supplementary Discussion 1 and 2 as part of the main manuscript as the reviewer suggested (**Page 35, Lines 822-826; Page 36, Lines 837-847**). Supplementary Discussion 3 was retained since it is more like a supplementary note for the fly experiment. However, we are open to the option of including this analysis as part of the main manuscript if the reviewer thinks this is best.

References

1. Liu, J. *et al.* Sequence- and structure-selective mRNA m(5)C methylation by NSUN6 in animals. *Natl Sci Rev* **8**, nwaa273 (2021).
2. Xiao, Y. *et al.* An Elongation- and Ligation-Based qPCR Amplification Method for the Radiolabeling-Free Detection of Locus-Specific N(6) - Methyladenosine Modification. *Angew Chem Int Ed Engl* **57**, 15995-16000 (2018).
3. Begik, O. *et al.* Quantitative profiling of pseudouridylation dynamics in native RNAs with nanopore sequencing. *Nat Biotechnol* **39**, 1278-1291 (2021).
4. Zhang, H. *et al.* Dynamic landscape and evolution of m6A methylation in human. *Nucleic Acids Res* **48**, 6251-6264 (2020).

REVIEWERS' COMMENTS

Reviewer #1 (Remarks to the Author):

The authors have done a very conscientious job of complying with all the criticisms/suggestions of the referees. I recommend that the paper be published. Thank you for the impressive jobs and efforts.

Reviewer #2 (Remarks to the Author):

Unfortunately, the authors were not able to address my key point which is to use a biochemical method to establish the exact stoichiometry of m5C. I think this would have substantially enhanced the manuscript. They have successfully addressed other more minor issues. I still think this is an important addition and appropriate for Nature Communications though I wish precise stoichiometries modification could have been measured.

Reviewer #3 (Remarks to the Author):

I am satisfied that the authors have addressed the great majority of concerns raised by the first round reviews.